# Ripples reflect a spectrum of synchronous spiking activity in human anterior temporal lobe

Ai Phuong S Tong[1], Alex P Vaz[2], John H Wittig[1], Sara K Inati[3], Kareem A Zaghloul[1]*

[1]Surgical Neurology Branch, National Institute of Neurological Disorders and Stroke, National Institutes of Health, Bethesda, United States; [2]Medical Scientist Training Program, Duke University School of Medicine, Durham, United States; [3]Office of the Clinical Director, National Institute of Neurological Disorders and Stroke, National Institutes of Health, Bethesda, United States

**Abstract** Direct brain recordings have provided important insights into how high-frequency activity captured through intracranial EEG (iEEG) supports human memory retrieval. The extent to which such activity is comprised of transient fluctuations that reflect the dynamic coordination of underlying neurons, however, remains unclear. Here, we simultaneously record iEEG, local field potential (LFP), and single unit activity in the human temporal cortex. We demonstrate that fast oscillations within the previously identified 80–120 Hz ripple band contribute to broadband high-frequency activity in the human cortex. These ripple oscillations exhibit a spectrum of amplitudes and durations related to the amount of underlying neuronal spiking. Ripples in the macro-scale iEEG are related to the number and synchrony of ripples in the micro-scale LFP, which in turn are related to the synchrony of neuronal spiking. Our data suggest that neural activity in the human temporal lobe is organized into transient bouts of ripple oscillations that reflect underlying bursts of spiking activity.

*For correspondence:
kareem.zaghloul@nih.gov

**Competing interest:** The authors declare that no competing interests exist.

## Introduction

A fundamental premise in interpreting the various fluctuations and temporal dynamics observed in direct recordings from the human brain is that these signals must be related to the underlying synaptic currents and spiking activity of individual neurons (*Buzsáki et al., 2012*; *Parvizi and Kastner, 2018*). Arguably, the most robust link between intracranial EEG (iEEG) signals and neuronal activity has been that increases in broadband high-frequency activity are associated with overall increases in underlying neuronal spiking (*Manning et al., 2009*; *Burke et al., 2015*). This relation has shaped the insights we have gained regarding the neural substrates of human memory (*Jacobs and Kahana, 2010*). Successful episodic memory formation, for example, is accompanied by increases in broadband activity that progress anteriorly along the temporal cortex, which has consequently suggested that successful memory involves increases in neuronal spiking in these regions (*Burke et al., 2014*; *Long et al., 2014*; *Greenberg et al., 2015*).

The relation between widespread and prolonged increases in broadband high-frequency activity and successful memory formation has largely rested upon averaging neural data over multiple similar trials or events. This approach, however, obscures the moment to moment fluctuations that can arise as individuals try to encode or retrieve individual memories. Individual trials often exhibit increases in oscillatory and broadband activity that can be quite transient, as has been observed in recent studies of working memory (*Jones, 2016*; *Lundqvist et al., 2016*). Given the relation between broadband

power and neuronal spiking, these short bouts of broadband activity may reflect brief bursts of population spiking activity. Bursts of spiking are in fact common in cortical recordings in animals and may represent packets of information that are used as the building blocks for neural coding in the brain (*Luczak et al., 2009*; *Luczak et al., 2015*). The possibility that punctate events observed in the cortical iEEG signal may reflect underlying packets of spiking, however, has not been well explored in the human brain.

A parallel and extensive line of research, however, has explicitly demonstrated the presence of discrete bouts of fast oscillations known as ripples that have been identified using smaller scale local field potential (LFP) recordings in the rodent medial temporal lobe (MTL) in studies of spatial navigation (*Colgin, 2016*). These ripples are strongly associated with underlying bursts of spiking activity (*Buzsáki, 2015*). Ripples have been implicated in memory formation, consolidation, and retrieval (*Buzsáki, 2015*; *Joo and Frank, 2018*) and the bursts of spiking activity that accompany ripples are often organized into specific temporal sequences that have been hypothesized to represent the content of memory (*Carr et al., 2011*; *Vaz et al., 2020*; *Pfeiffer, 2020*). Recent reports have identified similar fast oscillations in the human brain even at larger spatial scales, and have suggested that these may be analogous to the ripples identified in rodents (*Axmacher et al., 2008*; *Zhang et al., 2018*; *Vaz et al., 2019*; *Norman et al., 2019*). Moreover, fast oscillations that appear similar to MTL ripples can also be identified in the cortex both in animals and in humans (*Khodagholy et al., 2017*; *Vaz et al., 2019*). Whether these cortical ripples should be considered the same as ripples in the MTL is still a matter of debate, although recent evidence has demonstrated that such cortical ripples observed in the human cortex are similarly accompanied by underlying bursts of spiking (*Vaz et al., 2020*).

One of the challenges in resolving the nature of these fast oscillations that are observed in the human cortex and that appear similar to ripples observed in the MTL, however, is that ripple characteristics themselves can vary significantly across brain areas, behavioral states, and arousal levels (*Buzsáki, 2015*). Ripples likely do not exist as static entities, and behaviorally relevant changes in ripple characteristics have already been observed in humans (*Ngo et al., 2020*). This ambiguity of ripple morphology, especially during the awake state, is reflected in the variety of approaches used to identify ripples in both rodents and humans, in both the hippocampus and cortex (*Axmacher et al., 2008*; *Staresina et al., 2015*; *Buzsáki, 2015*; *Vaz et al., 2019*; *Jiang et al., 2020*). The variability in the amplitude and duration of ripples often makes it unclear whether any one event should be classified as a ripple, how to systematically identify thresholds for detecting them, and how to distinguish these discrete events from background activity.

Instead, the morphological features of ripples more likely exist on a continuum that reflects the activity and interactions among underlying neurons. Ripples depend on the extent of oscillatory coupling between pyramidal neurons and interneurons (*Csicsvari et al., 1999*; *Stark et al., 2014*), which can also change based on brain state and can differ between species and across brain regions (*Klausberger et al., 2003*). Cortical ripples in rodents exist on a spectrum of amplitudes that are highly correlated with underlying spiking activity (*Khodagholy et al., 2017*). Hence, a more direct approach for determining whether ripple oscillations identified in human cortical iEEG recordings might be functionally meaningful is to explicitly link the presence and characteristics of these observed cortical ripples with underlying spiking activity.

Here, we recorded macro-scale iEEG, micro-scale LFP, and single unit spiking activity in the human temporal lobe in order to examine the relation between cortical ripples and underlying neuronal spiking. We find that a major contributor to the changes in high-frequency power observed with successful memory retrieval are temporally punctate ripple events. These ripples exist on a spectrum of amplitudes and durations that are related to the extent of underlying spiking activity. The amplitude of ripples in the larger scale iEEG is related to the extent of synchronization across the underlying micro-scale LFP ripple oscillations, and neuronal spiking is locked to the trough of each ripple at the micro-scale. Together, our data suggest that many of the changes in broadband high-frequency power observed in direct recordings of the human brain during cognition may reflect ripple events.

## Results

### High-frequency activity reflects discrete 80-120 Hz ripples

We examined intracranial EEG (iEEG) recordings in 21 participants with intracranial electrodes placed for seizure monitoring as they performed a verbal episodic memory task (*Figure 1A*; see Materials and methods). In an example electrode in the medial temporal lobe, we observed transient increases in high-frequency activity (70–200 Hz; HFA) in individual trials immediately before participants vocalize their response during the retrieval period (*Figure 1C*). When averaging across all trials, the increases in HFA prior to vocalization appear sustained, consistent with previous studies of episodic memory retrieval (*Burke et al., 2014*; *Yaffe et al., 2014*).

We hypothesized that the transient increases in HFA observed in individual trials may be related to discrete 80–120 Hz ripples that have been previously associated with human memory retrieval (*Vaz et al., 2019*). We therefore identified ripples in each iEEG electrode in each participant (ripple rate .35 ± .04 Hz (mean ± SEM) across electrodes across all participants; *Figure 1B*; *Figure 1—figure supplement 1* to *Figure 1—figure supplement 5*; see Materials and methods). In the same example electrode, the transient increases in HFA observed in each trial correspond to the detection of individual ripples (*Figure 1C*). Across all trials, ripple rates demonstrate a clear increase that coincides with the sustained increase in HFA. We examined whether the changes in HFA and ripple rate were similarly modulated by successful memory retrieval in this examplar electrode (*Figure 1D*). Both HFA and ripple rates increase prior to vocalization during successful memory retrieval trials compared to trials in which the participant failed to successfully retrieve the correct word.

Since ripple characteristics can vary significantly across brain regions, we examined whether the differences in HFA and ripple rate between correct and incorrect retrieval trials exhibit a similar spatial pattern across the brain. Across participants, HFA increases during the one second prior to vocalization are greater during correct memory retrieval compared to incorrect memory retrieval in several anatomic regions (*Figure 1E*; see Materials and methods). When examining ripple rates, we also found significant increases during correct compared to incorrect trials in similar anatomic regions during this same time period (*Figure 1E*; *Figure 1—figure supplement 6B*). Across ROIs from the entire cortex, the participant average difference in HFA between correct and incorrect trials is positively correlated with the difference in ripple rate ($r = .13$, $p = 6.1 \times 10^{-4}$; *Figure 1F*). Within two specific brain regions, the medial temporal lobe and the anterior temporal lobe, this positive correlation between the participant average difference in HFA between correct and incorrect trials and the average difference in the ripple rate is greater (medial temporal lobe, MTL: $r = .60$, $p = .00072$; anterior temporal lobe, ATL: $r = .23$, $p = .026$; *Figure 1G*; *Figure 1—figure supplement 6C,D*). Together, these data suggest that the changes in HFA and ripple rate observed with successful memory retrieval obey a similar anatomic distribution.

We then examined this relation between HFA and 80–120 Hz ripples within retrieval trials in all individual electrodes in all participants. In each electrode, we computed the Pearson correlation between the average HFA and ripple rate across all retrieval trials, performed a Fisher's $z$-transform to normalize the correlation coefficients across participants, and then computed an average across all electrodes in each participant. We similarly computed this correlation across random three second epochs throughout the experimental session, which we designated as baseline. In both cases, the distribution of correlations across participants was consistent and significantly greater than zero (baseline $r = .085 ± .018$; retrieval $r = .093 ± .020$; $t(20) = 4.6$, $p < 1 \times 10^{-6}$, one-tailed t-test; *Figure 1H*). The relation between HFA and ripple rate was similar during memory retrieval and baseline (retrieval-baseline $.0079 ± .0016$, $t(20) = 1.44$, $p = 0.083$, paired t-test). However, the relation was stronger during correct retrieval as compared to incorrect retrieval (correct $r = .316 ± .0339$, incorrect $r = .259 ± .0296$; $t(20) = 8.7$, $p < 2 \times 10^{-8}$; correct-incorrect $.057 ± .004$, $t(20) = 2.40$, $p = .013$, paired t-test; *Figure 1I*).

To explicitly examine the extent to which 80–120 Hz ripples contribute to 70–200 Hz power, we conducted a control analysis by removing the time indices in which ripples were detected from the iEEG trace and recomputed HFA power. While the distribution of 70–200 Hz power averaged across electrodes in each participant is significantly greater during correct compared to incorrect memory retrieval (correct-incorrect $.0051 ± .0049$, $t(20) = 2.39$, $p = .013$), removing the discrete 80–120 Hz ripples eliminates this difference (correct-incorrect: $.0075 ± .0053$, $t(20) = 1.42$, $p = .086$; *Figure 1J*; *Figure 1—figure supplement 6E*). HFA power during correct retrieval is significantly reduced when

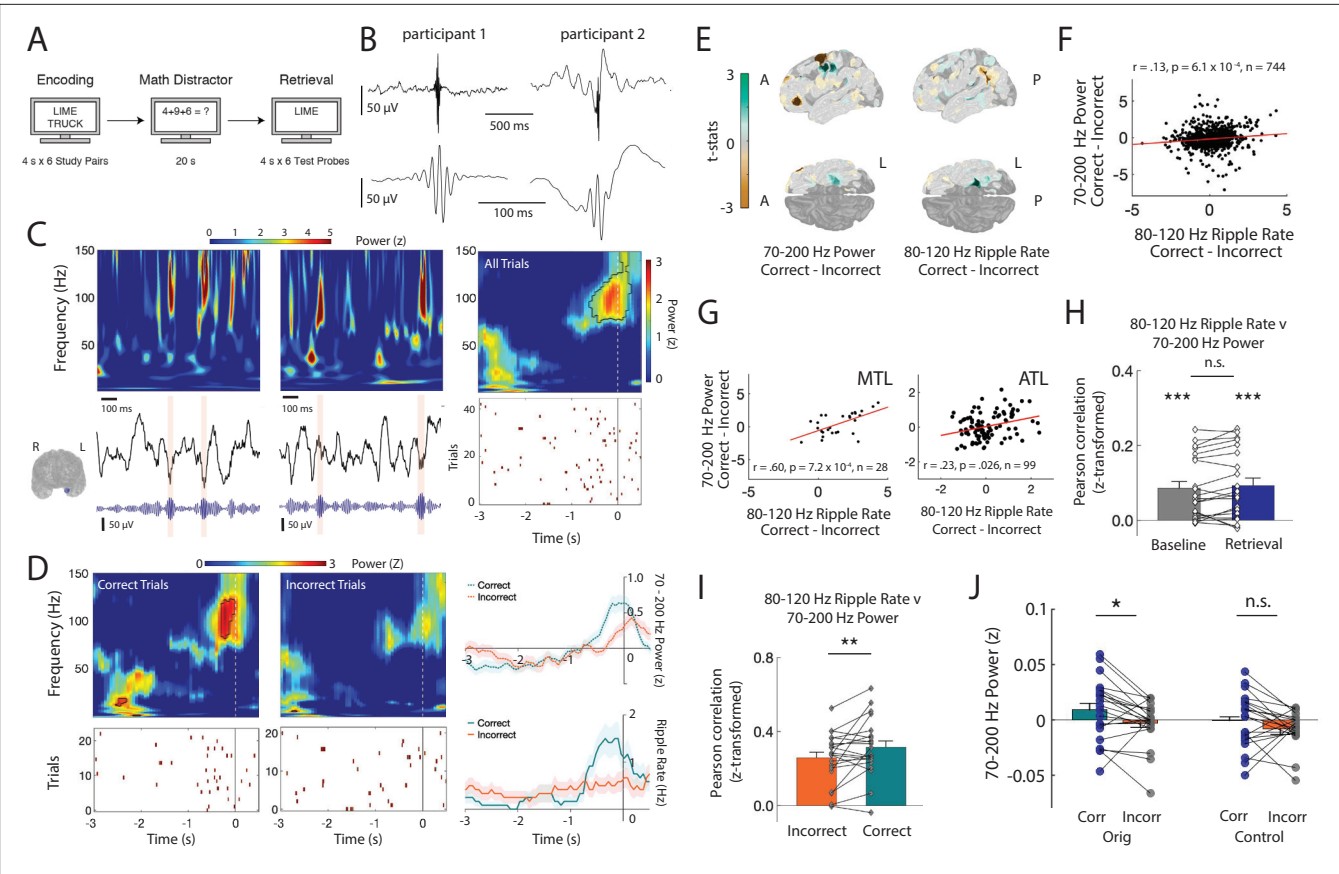

**Figure 1.** High-frequency activity reflects discrete ripples. (**A**) Paired-associates verbal episodic memory task. (**B**) Average iEEG signal locked to detected ripples in an anterior temporal lobe electrode in two participants. (**C**) Time-frequency power spectrograms for two clips of iEEG data from one electrode in medial temporal lobe (MTL) with corresponding iEEG voltage signal (black), 80–120 Hz band signal (blue), and detected ripple events (shaded). Location of the representative channel is shown. Trial-averaged power spectrogram of the single channel in medial temporal lobe during retrieval (top right) and corresponding spike raster of iEEG ripples across trials prior to vocalization (bottom right). Black contour indicates significant clusters (cluster-based permutation, p < .01). (**D**) Trial-averaged power spectrograms and corresponding ripple raster plots for correct and incorrect retrieval. Average 70–200 Hz power time series (top right) and average ripple rate time series (bottom right) for correct and incorrect retrieval. Black contour indicates significant clusters (cluster-based permutation, p < .01). (**E**) Cortical topographic plots of difference in 70-200 Hz power and 80-120 Hz ripple rate between correct and incorrect memory retrieval. Each data point reflects the across-participant t-statistic for a region of interest (ROI). (**F**) Pearson correlation between 70-200 Hz power and 80-120 Hz ripple rate across all ROIs. Each data point represents the average across participants for each ROI. Line represents the least-squares regression. (**G**) Pearson correlation between 70–200 Hz power and 80–120 Hz ripple rate across all ROIs in the medial temporal lobe (MTL) and anterior temporal lobe (ATL). Lines represent least-squares regression. (**H**) Fisher z-transformed Pearson correlation between 70–200 Hz power and 80–120 Hz ripple rate across all electrodes at baseline and during memory retrieval. Each participant is represented by a data point (*** p < .001). (**I**) Fisher z-transformed Pearson correlation between 70–200 Hz power and 80–120 Hz ripple rate across all electrodes during correct and incorrect memory retrieval (* p < .05). (**J**) Average 70–200 Hz power across all electrodes during correct compared to incorrect memory retrieval in true data (*Orig*) and after removal of the temporal indices of detected ripples (*Control*); (*** p< .001; * p < .05). Code and data is provided in *Figure 1—source code 1* and at https://doi.org/10.5061/dryad.5qfttdz6t.

The online version of this article includes the following source code and figure supplement(s) for figure 1:

**Source code 1.** Matlab code of ripple events in the iEEG signal.

**Figure supplement 1.** Ripple-triggered average iEEG and LFP signals.

**Figure supplement 2.** iEEG and LFP ripple characteristics with different detection thresholds.

**Figure supplement 3.** Multiple oscillations detection algorithm detected narrowband oscillations.

**Figure supplement 4.** MTL-ATL cross-correlograms with different detection thresholds.

**Figure supplement 5.** Interictal epileptiform discharge detection and overlap with ripples.

**Figure supplement 6.** High-frequency activity reflects discrete ripples.

removing the 80–120 Hz ripples (original correct-control correct: .0096 ± .0023, $t(20)$ = 4.21, p = 2.2 × $10^{-4}$). We also examined the correlation across all ROIs between the participant average difference in ripple rate between correct and incorrect trials and the difference in HFA after removal of the discrete ripples and found that this relation is no longer significant ($r$ = -.042, p = .270; *Figure 1—figure supplement 6F*,G). Finally, to confirm that much of the 70–200 Hz power is driven by relatively band limited 80–120 Hz ripples, we repeated our analysis after detecting discrete ripple events in a higher 120–200 Hz frequency band. Across ROIs from the entire cortex, we did not find a significant correlation between the participant average difference in HFA between correct and incorrect trials and the difference in 120–200 Hz ripple rate (*Figure 1—figure supplement 6H*).

## Ripple band amplitudes reflect a spectrum of underlying local spiking activity

In a subset of six participants, we had the opportunity to record micro-scale local field potentials (LFPs) and single unit spiking activity from a micro-electrode array (MEA) implanted in the anterior lateral temporal lobe underneath the iEEG electrodes (*Figure 2A*; *Figure 2—figure supplement 1*; see Materials and methods). In an example participant, ripples present in a single iEEG recording electrode overlying the MEA clearly occur simultaneously with ripples in the LFP across multiple micro-electrodes within the MEA (*Figure 2B*; *Figure 2—figure supplement 2*). These discrete events are accompanied by increases in spiking activity across multiple units, and therefore transient increases in the overall population spiking rate across the MEA (*Figure 2—figure supplement 3* and *Figure 2—figure supplement 4*).

The continuous time data of iEEG, LFP, and spiking activity suggest that 80–120 Hz ripples at both the macro- and micro- spatial scale and single unit spiking activity are related. To examine this relation, we computed $z$-scored 80–120 Hz ripple band amplitude in both the overlaying iEEG electrode and the average $z$-scored ripple band amplitude and spike rate in each of the MEA electrodes during 100 ms non-overlapping windows over all retrieval trials (ripple rate .84 ± .43 Hz across microelectrodes across all participants). Across all time windows in this participant, the average spike rate across MEA electrodes is significantly correlated with iEEG and LFP ripple band amplitude (spike rate v LFP amplitude $r$ = .61, p < 1 × $10^{-18}$; spike rate v iEEG amplitude $r$ = .12, Pearson correlation; *Figure 2C*). We found that the relation between spiking activity and ripple band amplitude at both spatial scales is consistent and significant across participants (spike rate v LFP amplitude, Fisher $z$-transform: $r$ = .751 ± .188; $t(5)$ = 4.00, one-tailed t-test; spike rate v iEEG amplitude: $r$ = .118 ± .049; $t(5)$ = 2.39, p = .031; *Figure 2D*).

These data demonstrate that the continuous time measure of 80–120 Hz ripple band amplitude is related to spiking activity. However, we were interested in understanding whether the amplitude and duration of discrete ripples may exist on a continuum reflecting underlying neuronal activity. We therefore relaxed our criteria for identifying discrete ripple events in order to detect ripples that are smaller and shorter duration, which are often assumed to be noise (see Materials and methods). We found ripples at both the macro- and micro-scale with a range of amplitudes and durations (*Figure 1—figure supplement 2*). During every ripple detected in each LFP trace, we collected the average spike rate of units recorded in the respective MEA electrode and computed the Pearson correlation between LFP ripple amplitude and spike rate across all ripples. Across participants, LFP ripple amplitude is consistently and significantly correlated with spike rate (Fisher $z$-transform, $r$ = .10 ± .02; $t(5)$ = 3.62, p = .008, one-tailed $t$-test; *Figure 2E*; *Figure 2—figure supplement 5A*). Even when ripples have amplitudes or durations below previously used thresholds, spiking activity is present in the microelectrode recording (*Figure 2—figure supplement 5B*).

While we found a strong relation between spiking activity and ripple amplitude, this observation could be confounded by a correlation between ripple amplitude and duration (*Figure 1—figure supplement 2E*). Larger amplitude are larger in duration and therefore may have more opportunity to co-occur with spikes by chance. To account for this, we shuffled the time indices of the detected spike times and computed the correlation between LFP ripple amplitude and the spike rate. The true relation between LFP ripple band amplitude and the spike rate is significantly greater than the shuffled distribution (true-shuffled $r$ = .096 ± .020; $t(5)$ = 3.312, p = .011, paired one-tailed $t$-test; *Figure 2F*).

In a similar manner, during every iEEG ripple we determined how many individual units spike as a proportion of the total number of units identified in each experimental session (*Figure 2—figure*

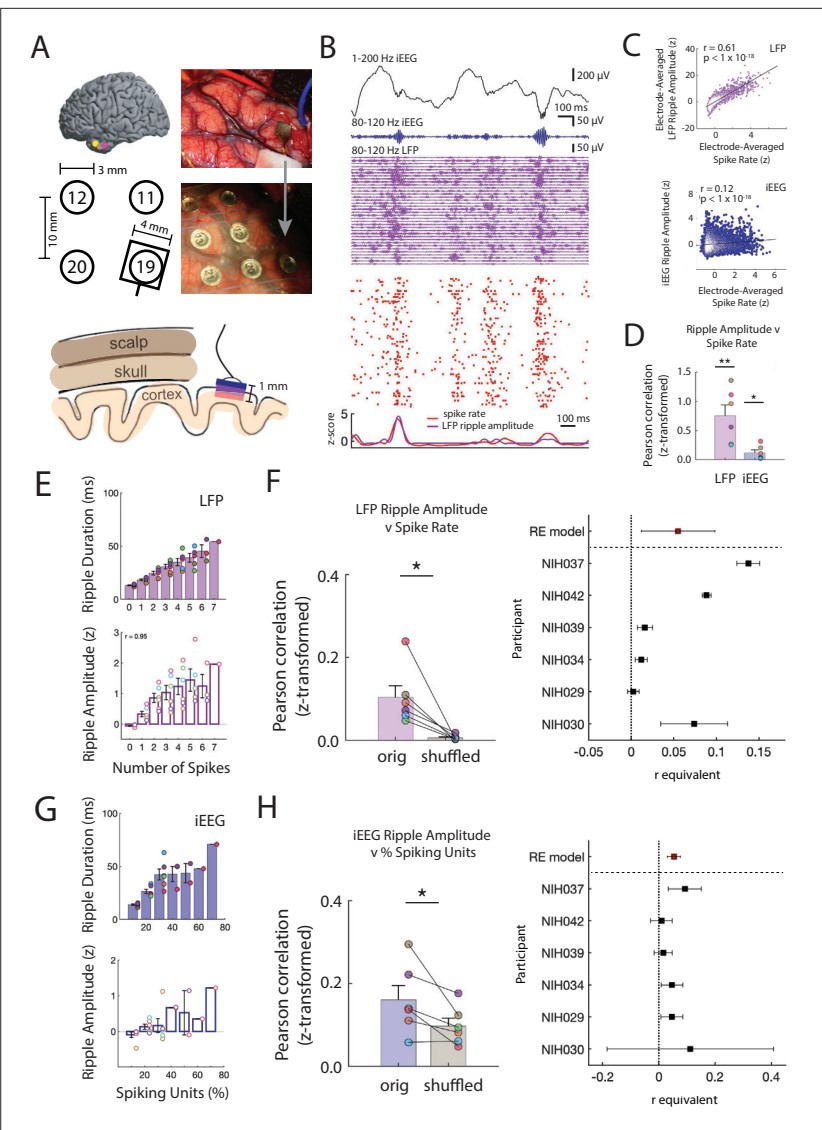

**Figure 2.** Ripple amplitudes reflect a spectrum of underlying local spiking activity. (**A**) Locations of the microelectrode arrays (MEA) in six participants (*top left*). Location of the MEA with respect to four nearby iEEG channels in one participant (*bottom left*). Intraoperative photo of implanted MEA in the ATL (*top right*) and after placement of an iEEG grid over the MEA (*bottom right*). Schematic of scalp, skull and cortex with respect to one iEEG channel on the cortical surface and one MEA that extends into cortex (*bottom*). (**B**) Brief 1500 ms window of 1-200 Hz iEEG signal (black), 80-120 Hz band iEEG signal (blue), 80-120 Hz band LFP signals across all MEA electrodes (purple), and spike raster for sorted units (red). (**C**) Pearson correlation between average spike rate and average LFP ripple amplitude across all MEA electrodes in one participant (blue). Pearson correlation between average spike rate and iEEG ripple band amplitude for one nearby iEEG electrode in one participant (purple). Each data point represents a 100 ms non-overlapping window. (**D**) Fisher *z*-transformed Pearson correlation between continuous spike rate and LFP and iEEG ripple amplitude. Group level statistics are shown as mean ± SEM across six participants. Each data point represents a participant (** p < .01, * p < .05). (**E**) Average duration and amplitude of ripples in the LFP signal related to the number of spikes during the ripple. Each data point represents a participant. (**F**) Fisher *z*-transformed Pearson correlation between spike rate and amplitude of coincident LFP ripple. Group level results are shown as mean ± SEM across six participants. Each data point represents a participant. True data (*orig*) compared to correlations when shuffling the spike time indices (*shuffled*; * p < .05). Forest plot of the r equivalent effect size and 95% CI for each participant and random-effect (RE) mean estimate across all participants (*right*). (**G**) Average duration and amplitude of ripples in the iEEG signal related to the number of spikes during the ripple. Each data point represents a participant. (**H**) Fisher *z*-transformed Pearson correlation between percentage of spiking units and amplitude of coincident iEEG ripple. Group level results are

*Figure 2 continued on next page*

*Figure 2 continued*

reported as mean ± SEM across participants. True data (*orig*) compared to correlations when shuffling the spike time indices (*shuffled*; * p < .05). Forest plot of the r equivalent effect size and 95% CI for each participant and random-effect (RE) mean estimate across all participants (*right*). Code and data is provided in ***Figure 2—source code 1*** and at https://doi.org/10.5061/dryad.5qfttdz6t.

The online version of this article includes the following source code and figure supplement(s) for figure 2:

**Source code 1.** Matlab code of correlations between continuous spiking, LFP and iEEG.

**Figure supplement 1.** MEA position with respect to iEEG channels.

**Figure supplement 2.** Raw iEEG and LFP Trace.

**Figure supplement 3.** Spiking auto-correlograms within and outside ripples.

**Figure supplement 4.** Ripple power and spike rate distributions.

**Figure supplement 5.** Ripples reflect underlying neuronal spiking.

---

*supplement 5C*). We computed the Pearson correlation between the percentage of actively firing units and the iEEG ripple band amplitude and duration across all detected iEEG ripples in every participant (***Figure 2H***; ***Figure 2—figure supplement 5D***). Across participants, this correlation is significant (Fisher $z$-transform, $r = .15 ± .03$; $t(5) = 4.679$, $p = .003$, one-tailed $t$-test). We performed the same shuffling procedure to account for ripple duration, and found that the true correlations across participants are significantly larger than the shuffled data (true-shuffled $r = .069 ± .027$; $t(5) = 2.517$, $p = .027$, paired one-tailed t-test; ***Figure 2H***). Together, these data demonstrate a strong relation between underlying unit spiking activity and ripples observed at the micro- and macro-scale in the human temporal cortex.

## Macro-scale ripples reflect number and alignment of micro-scale ripples

Our data suggest that ripples observed at both spatial scales may be related to one another. We hypothesized that the amplitude of the ripple observed in the iEEG signal is related to both the total number of LFP ripples and the extent to which the LFP ripples observed across the underlying MEA electrodes are aligned (***Figure 3A***).

We first examined the relation between the amplitude of the iEEG ripple and the number of LFP ripples simultaneously present in the MEA electrodes. In every participant, we detected ripples in each of the four iEEG electrodes closest to the MEA. For every detected ripple, we computed the mean 80–120 Hz ripple band amplitude across the iEEG electrodes and the number of LFP ripples simultaneously observed across the MEA (***Figure 3—figure supplement 1A***). In each participant, the iEEG ripple amplitude is positively correlated with the percent of MEA electrodes exhibiting LFP ripples across all detected iEEG ripples (% MEA electrodes with ripples, Fisher z-transform; $r = .11 ± .02$, $p = .005$; ***Figure 3B***; ***Figure 3—figure supplement 1B***,C). We accounted for the possibility that the longer durations observed in higher amplitude iEEG ripples may result in a larger number of detected LFP ripples by using a similar shuffling procedure. In this case, during each shuffle we performed a random circular shift of the time indices of the detected LFP ripples. After accounting for these longer durations, we still found that the true relation between iEEG ripple amplitude and the number of simultaneously detected LFP ripples is significantly greater than the shuffled distribution (true-shuffled $r = .05 ± .02$; $t(5) = 2.543$, $p = .026$, paired one-tailed t-test; ***Figure 3B***).

We then examined the relation between the amplitude of the iEEG ripple and the extent to which the LFP ripples in the underlying MEA are synchronized. For every detected iEEG ripple, we extracted the LFP 80–120 Hz ripple band instantaneous phase for all 96 MEA electrodes and computed the maximum pairwise phase consistency (PPC) over all time points within the duration of that iEEG ripple (***Figure 3C and D***; see Materials and methods). Across participants, the PPC is significantly correlated with the maximum amplitude of the observed iEEG ripples (Fisher z-transform; $r = 1.06 ± .43$; $t(5) = 2.34$, $p = .033$, one-tail t-test; ***Figure 3E–F***; ***Figure 3—figure supplement 1D***). In addition, the correlations across participants are significantly greater than those that would be observed by chance (true-shuffled $r = .03 ± .01$; $t(5) = 2.28$, $p = .036$, paired one-tail t-test; ***Figure 3E***; see Materials and methods). We repeated this analysis using only microelectrodes with detected ripples and found that, across participants, the PPC is still significantly correlated with the maximum amplitude of the observed iEEG ripples (Fisher z-transform; $r = .066 ± .02$; $t(5) = 3.71$, $p = .014$, one-tail t-test). These

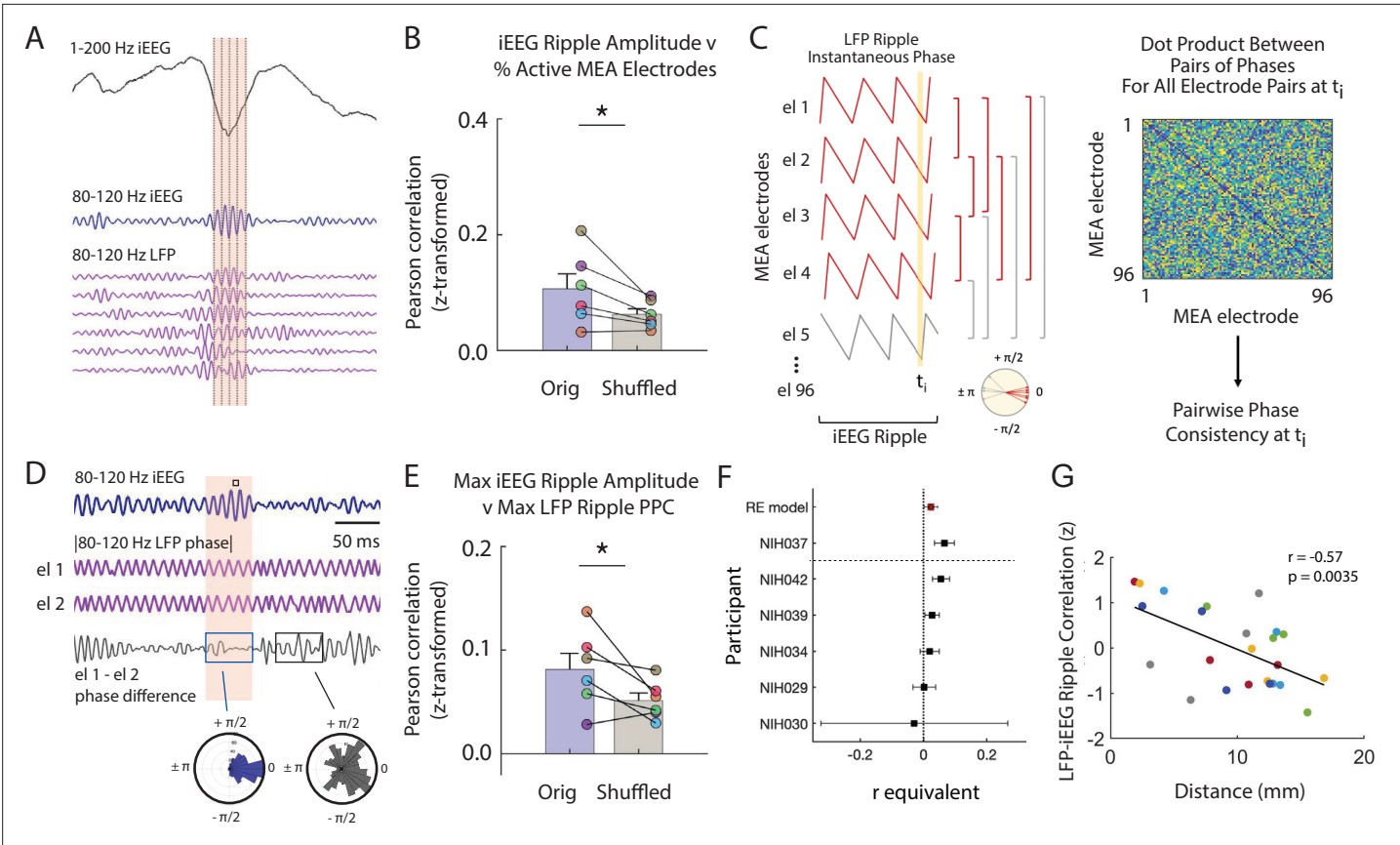

**Figure 3.** Macro-scale ripple amplitude reflects number and alignment of micro-scale ripples. (**A**) Brief window around one iEEG ripple showing unfiltered iEEG signal (black), ripple band iEEG signal (blue) and ripple LFPs for one nearby iEEG channel and six microelectrode array (MEA) electrodes with coincident LFP ripples. (**B**) Fisher *z*-transformed Pearson correlations for percentage of MEA electrodes containing LFP ripples and amplitude of coincident iEEG ripple. Group level results reported as mean ± SEM persists when duration of the iEEG ripple is accounted for by shuffling (* p < .05). Each data point represents a participant. (**C**) Schematic of calculation of pairwise phase differences across all microelectrodes to compute pairwise phase consistency. (**D**) Brief window around one iEEG ripple showing ripple band iEEG signal (blue), instantaneous phase of a pair of MEA electrodes (purple; out of many pairs, not shown) and instantaneous phase difference of the ch 1 and ch 2 pair (black). Maximum of iEEG ripple indicated with small black square in the shaded window above the ripple band iEEG signal. Polar histogram of all pairwise phase differences during a detected iEEG ripple is centered around 0 (blue). Polar histogram all pairwise phase differences outside of a iEEG ripple is more uniform (black). (**E**) Fisher *z*-transformed Pearson correlations between maximum pairwise phase consistency across all MEA electrode pairs and maximum amplitude of iEEG ripples. Group level results, reported as mean ± SEM, persists when duration of the iEEG ripple is accounted for by shuffling (* p < .05). Each data point represents a participant. (**F**) Forest plot of the r equivalent effect size and 95% CI for each participant and random-effect (RE) mean estimate across all participants. (**G**) Relation between distance between MEA and iEEG electrode and LFP-iEEG ripple synchrony. Each data point represents the relation between a MEA and iEEG electrode in the MTL or ATL, and each color represents a different patient. Code and data is provided in *Figure 3—source code 1* and at https://doi.org/10.5061/dryad.5qfttdz6t.

The online version of this article includes the following source code and figure supplement(s) for figure 3:

**Source code 1.** Matlab code of pairwise phase consistency between LFP ripple signal and iEEG ripple amplitude.

**Figure supplement 1.** Macro-scale ripple amplitude reflects number and alignment of micro-scale ripples.

**Figure supplement 2.** LFP-iEEG ripple cross-correlations for different detection thresholds.

**Figure supplement 3.** LFP-iEEG ripple cross-correlations with respect to distance.

data together suggest that the iEEG ripple reflects both the aggregate sum and alignment of the underlying LFP ripples.

To further examine the relation between ripples detected in the LFP signal and ripples detected in the iEEG explicitly, we measured the coincidence of ripples detected at the two spatial scales by computing the cross-correlogram of ripples detected in the LFP and iEEG traces. We found that ripples are coincident above chance for all detection parameters tested (*Figure 3—figure supplement 2*; see Materials and methods). Moreover, the extent to which ripples are coincident between

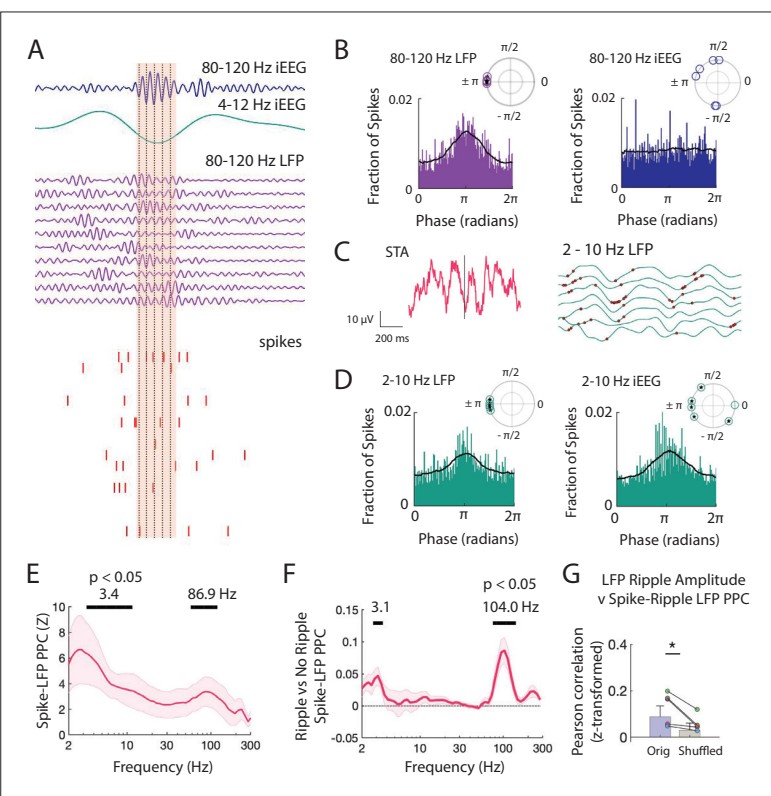

**Figure 4.** Spiking activity is phase-locked to ripples and low frequencies. (**A**) Brief window around one iEEG ripple and underlying LFP ripple and spiking activity. Dashed black lines indicate trough of iEEG ripple cycles compared to concurrent LFP ripple cycles and spiking. (**B**) Distribution of phases of LFP ripple (left) and iEEG ripple (right) across spike times for all units. (*inset*) Complex mean of the distribution of phases for each participant is depicted in a polar plot. Circles filled with a star if the distribution within a participant shows significant phase-locking (Rayleigh test, p < .001). Black line shows the average of six distributions across participants. (**C**) Spike triggered average (STA) for spikes detected within LFP ripples, in pink, with the 2-10 Hz bandpass filtered signal, in black. Bandpass filtered STA during correct (green) and incorrect (orange) trials are also shown. (*right*), Brief 500 ms window of 2-10 Hz filtered LFP (green) across MEA electrodes with neuronal activity. Red dots mark spikes occurring preferentially at trough of local LFP. (**D**) Distribution of phases of LFP low-frequency (left) and iEEG low-frequency (right) signals across spike times for all units. (*inset*) Complex mean of the distribution of phases for each participant is depicted in a polar plot. Circles filled with a star if the distribution within a participant shows significant phase-locking (Rayleigh test, p < .001). Black line shows the average of six distributions across participants. (**E**) Mean ± SEM spike-LFP PPC across participants for all spikes to LFP for every frequency between 2 and 300 Hz. Peak frequencies of significant clusters are shown. (**F**) Mean ± SEM difference in spike-LFP PPC between spikes that co-occur with LFP ripples and spikes that do not across participants. Peak frequencies of significant clusters are shown. (**G**) Fisher *z*-transformed Pearson correlations between spike-LFP ripple PPC and LFP ripple amplitude. Each data point represents a participant. True data (*orig*) compared to correlations when shuffling the spike time indices (*shuffled*; * p < .05). Code and data is provided in *Figure 4—source code 1* and at https://doi.org/10.5061/dryad.5qfttdz6t.

The online version of this article includes the following source code and figure supplement(s) for figure 4:

**Source code 1.** Matlab code of pairwise phase consistency between spiking and LFP.

**Figure supplement 1.** Spiking activity is phase-locked to ripples and low frequencies.

**Figure supplement 2.** Spike-LFP PPC for different ripple detection thresholds.

electrodes at the two spatial scales is significantly related to the distance between them (all participants: $r$ = -.572, p = .0035; across each participant, n = 6: $r$ = -.595 ± .271, mean ± SEM, $t$(5) = -2.40, p = .0615; *Figure 3G*; *Figure 3—figure supplement 3*).

## Spiking activity is phase-locked to ripples

Given that synchronization of LFP ripples was associated with increased ripple amplitudes, we also hypothesized that spike timing during ripples would be phase locked (*Le Van Quyen et al., 2008*). In individual participants, we often observed that unit firing preferentially occurs at the trough of the corresponding LFP ripples (*Figure 4A*). When the LFP ripples are aligned, spiking activity also appears to preferentially occur at the trough of the overlying iEEG ripple. In all participants, spikes from all units are locked to the trough of the 80–120 Hz ripple band in the LFP signal (p < $10^{-4}$, Rayleigh test across all units in each participant; p = $4.8 \times 10^{-4}$, Rayleigh test across six complex means, one from each participant; *Figure 4B*). We did not observe such phase consistency when examining the extent to which spiking activity is locked to the phase of 80–120 Hz ripple band activity in the macro-scale iEEG signal (p = .12, Rayleigh test across all units and across participants; *Figure 4B*).

When we visualized the spike triggered average of the LFP signal in individual participants, we often observed that spiking activity also appeared locked to negative deflections in the LFP (*Figure 4C*). These negative deflections contain spectral power within low frequencies. We therefore also examined the distribution of 2–10 Hz low-frequency phases present in the LFP signal around each spike and found significant locking to the trough in all participants (p < $10^{-4}$, Rayleigh test; p = $5.4 \times 10^{-4}$, Rayleigh test across six complex means, one from each participant; *Figure 4D*). Spikes from all units appear locked around the trough of the 2–10 Hz low-frequency iEEG signal when pooled, which reflects the negative deflection in the iEEG signal, but when examined separately for each participant, the apparent spike locking to the 2–10 Hz iEEG signal is not consistently at the same phase across participants (p < $10^{-4}$, Rayleigh test across all units in each participant; p = .70, Rayleigh test across six complex means, one from each participant; *Figure 4D*).

To examine the relation between spiking activity and individual frequencies within the LFP signal, we computed PPC across all spikes within each MEA electrode for each frequency between 2 Hz and 400 Hz (see Materials and methods) (*Vinck et al., 2010*). Across participants, spiking activity is significantly locked to specific high-frequency bands in the LFP (peak 86.9 Hz, p < .05, permutation test; see Materials and methods *Figure 4E*). We confirmed that spiking activity is locked to this high-frequency band across participants by also computing the phase-locking value (*Figure 4—figure supplement 1A*). This observed locking between spikes and this high-frequency band in the LFP signal was robust to different detection thresholds (*Figure 4—figure supplement 2*). Spikes also appear locked to a low-frequency band, but this likely represents the sharp negative deflections observed in the iEEG and LFP traces that accompany burst of spiking activity. Spikes are significantly more locked to high frequencies when they arise during ripples as compared to between ripples (p < .05, permutation test; *Figure 4*, *Figure 4—figure supplement 1B*).

We next examined the relation between the extent to which spiking activity locks to the 80–120 Hz frequency within each ripple and the amplitude of the ripple. Across participants, mean spike-LFP PPC within the 80–120 Hz ripple band is significantly correlated with 80–120 Hz ripple amplitude across all MEA electrodes (Fisher $z$-transform, $r$ = .084 ± .026, $t$(5) = 3.27, p = .011; *Figure 4G*). To account for any possible effects of ripple duration on the calculation of PPC, we compared this true distribution to a chance distribution and found that across participants LFP ripple amplitude exhibits a significantly stronger correlation with spike locking to the 80–120 Hz band in the true data as compared to the chance distribution ($t$(5) = 2.64, p = .023; see Materials and methods). Together with our data examining the relation between spiking activity and ripple amplitude, these data suggest that the amplitude of ripples in the LFP signal may reflect both the sum and alignment of underlying spiking.

Finally, given the observed relation between spiking activity and ripples, we then examined whether ripples themselves also exhibit a phase preference. As with the individual spikes, we considered each LFP ripple as an event and visualized the ripple-triggered average of the iEEG and LFP signal (*Figure 4—figure supplement 1C*). Ripples appear to exhibit a clear relation with negative deflections in the iEEG and LFP trace. We therefore examined the distribution of 2–10 Hz low frequency phases present in the LFP signal during each LFP ripple and found significant locking to the trough in all participants (p < $10^{-4}$, Rayleigh test across all ripples in each participant; p = .0099, Rayleigh test

across complex means, one from each participant; *Figure 4—figure supplement 1E*). Micro-scale ripples are also locked to the 2–10 Hz low-frequency band in the iEEG signal within individual participants but at variable phases ($p < 10^{-4}$, Rayleigh test across all ripples in each participant; $p = .72$, Rayleigh test across six complex means, one from each participant).

## Discussion

Despite significant advances over the past several decades, how to accurately interpret the various fluctuations and dynamics observed through direct recordings of the human brain has remained challenging. Our results demonstrate that many of the changes in broadband high-frequency activity captured through iEEG reflect transient 80–120 Hz oscillations. These short bouts of neuronal activity exist on a continuum of amplitudes and durations, and reflect underlying bursts of neuronal spiking.

We consider the possibility that these brief neuronal events are ripple oscillations that may be contributing to human cognition. One of the challenges, however, in examining the role of ripple oscillations in cognition, especially in the human brain, has been determining whether any particular event does or does not qualify as a ripple. Many of the criteria used for defining ripples in human recordings have been drawn from the more developed literature examining ripple oscillations in the rodent MTL (*Buzsáki, 2015*; *Joo and Frank, 2018*). Researchers interested in studying ripples, whether in the cortex or in the MTL, often choose fixed parameters based on these previous studies. However, fixed criteria may not accommodate the reality that ripples are dynamic entities with morphologies that can vary based on brain region or behavior (*Buzsáki, 2015*; *Ngo et al., 2020*). Moreover, it is not clear how these parameters that have been well established in rodents translate across different species, as ripples in human brain recordings for example have only been relatively recently described (*Axmacher et al., 2008*; *Staresina et al., 2015*; *Vaz et al., 2019*; *Norman et al., 2019*; *Jiang et al., 2020*; *Norman et al., 2021*).

Our data demonstrate that cortical ripples captured through human brain recordings exist on a continuum of amplitudes and durations. Our results do not prescribe a fixed set of criteria for identifying ripples, but instead highlight the point that strictly adhering to predefined criteria for what constitutes a ripple may run the risk of overlooking functionally meaningful events. Indeed, we explicitly explore this point here by using more liberal thresholds for ripple detection. By recording neural activity across spatial scales, we find that even ripples with smaller amplitudes or shorter durations are associated with discrete bursts of spiking activity. The amplitude and duration of each ripple in the micro-scale LFP signal is related to the amount of neuronal spiking activity and the extent to which such spiking is synchronous. In turn, the amplitude and duration of each ripple in the macro-scale iEEG recording is related to the number and synchrony of ripples at the micro-scale. These results are consistent with previous studies of ripples conducted through both in vivo and slice recordings of rodent MTL structures which have suggested that ripples reflect the synchronous interactions and overall activity of underlying neurons (*Csicsvari et al., 1999*; *Stark et al., 2014*; *Khodagholy et al., 2017*; *Nitzan et al., 2020*). Although the durations of ripples we observe in our human recordings are shorter than those observed in the rodent MTL, this could be related to differences in the neural architecture, and therefore differences in the latencies of activation among individual neurons, between species or between brain regions.

The discovery that such transient bouts of narrow band oscillatory activity may be functionally relevant, both in the human MTL but also in the human cortex, has raised the possibility that these events are similar to MTL ripples that have been extensively described in rodents (*Axmacher et al., 2008*; *Staresina et al., 2015*; *Buzsáki, 2015*; *Vaz et al., 2019*; *Jiang et al., 2020*; *Norman et al., 2021*). Whether ripples are specific to the MTL or whether they are a more general feature of neural processing is still a matter of debate. Our data demonstrate similar events in the human cortex, fast oscillations within a narrow 80–120 Hz band of activity that we identify using multiple complementary analyses. We excluded the possibility that these events are related to epilepsy and interictal epileptiform discharges, and we find that these events are associated with ripples in the MTL. Importantly, these events are related to bursts of underlying spiking activity. We consequently label them as ripples given their similarity and relation with MTL ripples. Regardless of their exact label, however, these events appear to reflect transient bouts of spiking activity that are related to information processing in the brain.

Our work is also consistent with several prior studies demonstrating a strong association between gamma power, broadband high frequency power, and spiking activity (*Berens et al., 2008*; *Manning et al., 2009*; *Panagiotaropoulos et al., 2012*; *Burke et al., 2014*). We similarly find a strong relation between spiking activity and ripples, which in our analyses occupy a narrow band of frequencies between 80–120 Hz. It is possible that these phenomena are related, and that the previously described gamma band or broadband activity simply includes this narrow ripple band. We find that this narrow band activity accounts for many of the changes observed in the broadband power, and, of note, the cortical spiking activity in our data is locked to this narrow band.

By examining both iEEG and LFP recordings in the human brain for the presence of discrete ripples, our data therefore support the hypothesis that many of the dynamics observed in broadband high frequency activity captured from the human brain are driven by well-defined and brief bouts of neural oscillatory activity that reflect bursts of synchronized spiking. A common approach for investigating the neural correlates of human cognition has been to average neural activity over multiple trials and over broad frequency ranges (*Burke et al., 2014*; *Burke et al., 2015*; *Long et al., 2014*; *Greenberg et al., 2015*; *Wittig et al., 2018*). This approach has guided our understanding of human episodic memory formation, for example, but fails to account for the possibility that the neural mechanisms of memory may be more punctate (*Jones, 2016*; *Lundqvist et al., 2016*). The relation between band limited 80–120 Hz ripples and broadband high-frequency activity that we observe in our data suggests that many of the interpretations regarding the neural substrates of human memory may be better served by considering these transient events. It is important to recognize, however, that this relation is not absolute and appears less robust outside of the MTL and ATL. Even within these brain regions, this relation is clearer only during correct compared to incorrect memory retrieval. Hence, while 80–120 Hz ripples may underlie many of the phenomena observed through broadband high-frequency activity, there are likely other neural mechanisms that contribute to the dynamics observed in the iEEG signal.

The possibility that information is neurally encoded through packets of activity has been relatively under-explored in human brain recordings. Recent evidence captured through animal recordings related to both memory and perception, however, supports this possibility (*Lundqvist et al., 2016*; *Luczak et al., 2009*; *Luczak et al., 2015*). These advances are partly due to the more sophisticated tools that are available for in vivo recordings of large populations of spiking neurons in animals. By recording spiking activity from a population of neurons in the human temporal cortex through microelectrode arrays, we find direct evidence that 80–120 Hz ripples that we observe in our data are accompanied by bursts of neuronal spiking. Hence, our data demonstrate that neural activity in the human temporal cortex may be temporally organized into discrete bursts of spiking. Our data focus on these bursts of spiking activity as participants form and retrieve memories since the information contained within these bursts has been recently linked with memory retrieval (*Vaz et al., 2020*; *Pfeiffer, 2020*). However, our data cannot address whether the relation between ripples and underlying bursts of synchronized spiking is unique to just the temporal lobe or just to memory. Ripples have been most studied in the MTL in both animals and humans, but appear to jointly occur in brain regions that either process or receive the same information (*Lisman and Jensen, 2013*; *Khodagholy et al., 2017*; *Vaz et al., 2019*; *Swanson et al., 2020*). It is possible that this relation between ripples and spiking activity is unique to brain regions that communicate directly with the MTL or that are directly involved in memory. Our results, however, raise the possibility that such packets may be a general feature of neural coding in the human brain.

Given previous evidence demonstrating that spiking activity within ripples appears locked to the trough of each cycle, it is not surprising that we observe similar locking in our data (*Le Van Quyen et al., 2008*; *Nitzan et al., 2020*). We find more consistent locking of spiking activity to higher frequencies in the micro- compared to macro-scale. This may be because synchronous spiking can occur within local neuronal ensembles while varying across ensembles. However, we also find that spikes, and consequently ripples themselves, appear to coincide with large deflections in the iEEG and LFP trace that appear to have spectral power within a low-frequency band. Such locking of both spiking activity and ripples to the trough of these deflections can account for several phenomena that have been previously described in human brain recordings. For example, phase amplitude coupling between low frequency oscillations and high-frequency activity is ubiquitous in human recordings and has been linked to behavior (*Canolty et al., 2006*; *He et al., 2010*; *Vaz et al., 2017*). If many of the increases in broadband high-frequency activity are related to ripples, then phase amplitude coupling

may emerge simply because ripples, and therefore spiking activity, coincide with large deflections, or sharp waves, in human brain recordings that reflect periods of concentrated synaptic inputs *Buzsáki, 2015*.

It is also possible that some of the locking we observe between low-frequency power and spiking, bursts of spiking, and therefore ripples, also reflects locking to true low-frequency oscillations. If so, then this could also suggest a possible mechanism by which bursts of spiking activity may be conveyed from one brain region to another. Oscillations observed in the iEEG have been hypothesized to facilitate communication between brain regions and modulate the excitability or timing of neuronal spiking (*Fries, 2015*; *Chapeton et al., 2019*). Indeed, low-frequency coherence may be related to successful memory formation (*Shirvalkar et al., 2010*; *Fell et al., 2011*; *Lega et al., 2012*). In this framework, these oscillations could open gates of communication, allowing the brain to convey a volley of neuronal spiking from one region to another. Recent evidence has also suggested that higher frequency oscillations that are synchronous across brain regions may also facilitate communication, although the evidence for this still remains unclear (*Bosman et al., 2012*; *Fries, 2015*; *Buzsáki, 2015*; *Ray and Maunsell, 2015*). Our data has implications for interpreting such higher frequency coherence, as two brain regions that each exhibit bursts of spiking activity, either conveyed from one to the other directly or driven by a third region, can each generate high frequency ripple oscillations. If the underlying neuronal interactions in each brain region are similar, the ripples may appear coherent and at the same high frequency. Conversely, if the underlying architecture of each region is different, then any resulting higher frequency oscillations may differ in morphology and frequency, and therefore appear desynchronized.

Together, our data offer insights into the dynamic fluctuations observed in direct recordings from the human brain and suggest that neural activity may be organized into discrete 80–120 Hz ripple events that reflect underlying bursts of neuronal spiking. Our data argue against using fixed criteria to identify these ripples, and instead demonstrate that these ripples exist on a continuum of activity. As each of these 80–120 Hz ripples reflects bursts of neuronal spiking with varying degrees of activity, our data more broadly suggest these ripple oscillations may constitute one of the primary substrates of human cognition.

## Materials and methods
### Participants
Twenty-one participants with drug resistant epilepsy underwent a surgical procedure in which platinum recording contacts were implanted on the cortical surface as well as within the brain parenchyma. In each case, the clinical team determined the placement of the contacts to localize epileptogenic regions. In all the participants investigated here, the clinical region of investigation was the temporal lobes.

For research purposes, in six of these participants (4 female; 34.8 ± 4.7 years old) we placed one or two 96-channel microelectrode arrays (MEA; 4 × 4 mm, Cereplex I; Blackrock Microsystems, Inc, Salt Lake City, UT) in the anterior temporal lobe (ATL) in addition to the subdural contacts. We implanted MEAs only in participants with a presurgical evaluation indicating clear seizure localization in the temporal lobe and the implant site in the ATL was chosen to fall within the expected resection area. Each MEA was placed in an area of cortex that appeared normal both on the pre-operative MRI and on visual inspection. Across participants, MEAs were implanted 14.6 ± 3.7 mm away from the closest subdural electrode with any ictal or interictal activity identified by the clinical team. Four out of the six participants received a surgical resection which includes the tissue where the MEAs were implanted. One participant had evidence of focal cortical seizure activity and received a localized resection posterior to the MEA site. One participant did not have a sufficient number of seizures during the monitoring period to justify a subsequent resection. Neither participant experienced a change in seizure type or frequency following the procedure, or experienced any noted change in cognitive function. The data captured from these MEAs in these participants were included in a previous study (*Vaz et al., 2020*).

Data were collected at the Clinical Center at the National Institutes of Health (NIH; Bethesda, MD). The Institutional Review Board (IRB) approved the research protocol (11 N-0051), and informed

consent was obtained from the participants and their guardians. All analyses were performed using custom built Matlab code (Natick, MA). Data are reported as mean ± SEM unless otherwise specified.

## Paired-associates memory task

Each participant performed a paired associates verbal memory task (*Yaffe et al., 2014*; *Jang et al., 2017*; *Vaz et al., 2020*). Previous studies have demonstrated that correct memory retrieval in this task is associated with increases in high-frequency activity (*Yaffe et al., 2014*; *Jang et al., 2017*; *Vaz et al., 2020*). Here, we replicate these previous findings using a subset of participants that were included in these previous studies as well as additional new participants. During the study period, participants were sequentially shown a list of word pairs and instructed to remember the novel associations between each pair of words (encoding). Later during testing, they were cued with one word from each pair selected at random and were instructed to say the associated word into a microphone (retrieval).

A single experimental session for each participant consisted of 25 lists, where each list contained six pairs of common nouns shown on the center of a laptop screen. The number of pairs in a list was kept constant for each participant. Words were chosen at random and without replacement from a pool of high-frequency nouns and were presented sequentially and appearing in capital letters at the center of the screen. We separated the study and test of each word pair by a minimum lag of two study or test items. During the study period, each word pair was preceded by an orientation stimulus ('+') that appeared on the screen for 250–300 ms followed by a blank interstimulus interval (ISI) between 500–750 ms. Word pairs were then presented stacked in the center of the screen for 4000 ms followed by a blank ISI of 1000 ms. Following the presentation of the list of word pairs, participants completed an arithmetic distractor task of the form A + B + C = ? for 20 seconds.

During the test period, one word was randomly chosen from each of the presented pairs and presented in random order, and the participant was asked to recall the other word from the pair by vocalizing a response. Each cue word was preceded by an orientation stimulus (a row of question marks) that appeared on the screen for 4000 ms followed by a blank ISI of 1000 ms. Participants could vocalize their response any time during the recall period after cue presentation. We manually designated each recorded response as correct, intrusion, or pass. A response was designated as pass when no vocalization was made, when the participants made an unintelligible vocalization like 'umm', or when the participant vocalized the word 'pass'. During pass trials where no vocalization was present, we assigned a response time by randomly drawing from the distribution of correct response time during that experimental session. We did not include such pass trials where no vocalization was present in our analysis of incorrect trials. We defined all intrusion and other pass trials as incorrect trials. A single experimental session contained 150 total word pairs. Each participant completed between 1 and 3 sessions (2.2 ± .3 per participant). Participants studied 93 ± 8 word pairs, and successfully recalled 30.1% ± 4.1% of words. While patients were presented with 150 words pairs in each experimental session, the number of word pairs they actually studied was reduced if they did not complete the session due to interruptions or participant fatigue.

## Intracranial EEG recordings

We collected intracranial EEG (iEEG) data from a total of 1660 subdural and depth recording contacts (79 ± 4 per participant; *Figure 1—figure supplement 6*). Subdural contacts were arranged in both grid and strip configurations with an inter-contact spacing of 10 mm. We captured iEEG signals sampled at 1000 Hz. For clinical visual inspection of the recording, signals were referenced to a common contact placed subcutaneously, on the scalp, or on the mastoid process. The recorded raw iEEG signals used for analyses were referenced to the system hardware reference, which was set by the recording amplifier (Nihon Kohden, Irvine CA) as the average of two intracranial electrode channels. We used the Chronux toolbox to apply a local detrending procedure to remove slow fluctuations (≤ 2 Hz) from the time series of each electrode and a regression-based approach to remove line noise at 60 Hz and 120 Hz (*Mitra and Bokil, 2009*). We did not see a noticeable peak at the 180 Hz harmonic when we surveyed the power spectral density of several electrodes for noise and therefore did not remove line noise at that harmonic to avoid introducing artifacts. We implemented additional thresholds to remove movement artifacts and pathological activity related to the patient's epilepsy.

We quantified spectral power and phase in the iEEG signals by convolving the voltage time series with 200 linearly spaced complex valued Morlet wavelets between 2 and 200 Hz (wavelet number 6).

We extracted data from all retrieval periods, beginning 4 s preceding vocalization to 1 s following vocalization and included a 1000 ms buffer on both sides of the clipped data. We squared and log-transformed the continuous-time wavelet transform to generate a continuous measure of instantaneous power for each frequency. To account for changes in power across experimental sessions, we z-scored power values separately for each frequency and for each session using the mean and standard deviation of all respective values for that session. When examining the average changes in high frequency activity (70–200 Hz) during memory retrieval across trials, we temporally smoothed the z-scored spectrogram for each iEEG channel using a sliding 600 ms window (90 % overlap) as a point of comparison with previous studies of human memory retrieval (*Greenberg et al., 2015*).

## Anatomic localization

We localized electrodes in each participant by identifying high-intensity voxels in a post-operative CT image, which was co-registered to a pre-operative T1-weighted MRI. Electrode locations were adjusted to account for routine post-operative parenchymal shift by applying a standardized algorithm combining intraoperative photography, electrode spatial arrangement, and dural and pial surface reconstructions (*Trotta et al., 2018*). Pial surfaces were reconstructed using FreeSurfer (http://surfer.nmr.mgh.harvard.edu) (*Fischl, 2012*) and were resampled and standardized using the AFNI SUMA package (*Cox, 1996*). The resulting surfaces each contained 198812 vertices per hemisphere, with vertices indexed in a standardized fashion, such that for any vertex $i$, the $i$ th vertex is located in an anatomically analogous location across participants. We identified the location of each MEA on each participant's surface reconstruction. We co-registered the individual participant reconstructions with a standard template brain, and visualized the locations of each participant's MEA on the template brain.

We aggregated vertices from the surface reconstruction into a standard set of surface-based regions of interest (ROIs) as previously described (*Figure 1—figure supplement 6*; *Trotta et al., 2018*). Briefly, we sampled 2400 equally spaced vertices per hemisphere to use as ROI centers. ROI centers were uniformly distributed across the surface at an average geodesic distance of approximately 5 mm. We assigned all vertices within a 10 mm geodesic radius of an ROI center to that ROI, which achieves a coverage of 99.9% coverage or greater of the pial surface in each participant (*Trotta et al., 2018*). Because ROIs overlap, vertices may be assigned to multiple ROIs. On average, there were 669.44 ± 74.30 vertices per ROI and each vertex mapped to 8.08 ± .90 ROIs. We modeled each electrode as a cylinder with radius 1.5 mm, found the pial vertices closest to it, and then assigned each electrode to the same ROIs as its nearest pial vertices. Due to the overlap between ROIs, each electrode is assigned to multiple ROIs and each ROI may contain more than one electrode. For analyses within ROIs across participants, we only included ROIs that contained electrodes from at least five participants.

## iEEG artifact removal

We implemented several measures to provide the most conservative sampling of non-pathological signals possible. We implemented a previously reported automated trial and electrode rejection procedure based on excessive kurtosis or variance of iEEG signals to exclude high-frequency activity associated with epileptiform activity (*Jang et al., 2017*; *Wittig et al., 2018*; *Vaz et al., 2019*). We calculated and sorted the mean iEEG voltage across all trials, and divided the distribution into quartiles. We identified trial outliers by setting a threshold, Q3 + w*(Q3-Q1), where Q1 and Q3 are the mean voltage boundaries of the first and third quartiles, respectively. We empirically determined the weight w to be 2.3. We excluded all trials with mean voltage that exceeded this threshold. The average percent removed across all sessions in each participant due to either system-level noise or transient epileptiform activity was 5.17% ± .86% of all electrodes and 2.89% ± .34% of all trials.

In addition, system level line noise, eye-blink artifacts, sharp transients, and inter-ictal epileptic discharges (IEDs) can confound the interpretation of our results. We therefore implemented a previously reported automated event-level artifact rejection (*Staresina et al., 2015*; *Vaz et al., 2019*). We calculated a z-score for every iEEG time point based on the gradient (first derivative) and amplitude after applying a 250 Hz high pass filter (for identification of IEDs). All time points within 100 ms of any time point that exceeded a z-score of 5 with either gradient or high-frequency amplitude were marked as artifactual. We visually inspected the resulting iEEG traces and found that the automated procedure reliably removed IEDs and other artifacts. In total, following exclusion of electrodes because

of artifact, we retained 1577 electrodes (75 ± 4 per participant) for analysis. We approximated a reference-free montage within each participant by subtracting the common average reference of all retained electrodes from the voltage trace of each individual electrode for that participant.

## Microelectrode recordings

In six participants, we additionally captured spiking activity and micro-scale local field potentials (LFP) from the MEAs implanted in the anterior temporal lobe. Microelectrodes were arranged in a 10 × 10 grid with each electrode spaced 400 µm apart and extending 1.5 mm into the cortical surface (1.0 mm for one participant). Post-operative paraffin blocks of the resected tissue demonstrated that the electrodes extended approximately halfway into the 3-mm-thick gray matter. We digitally recorded microelectrode signals at 30 kHz using the Cereplex I and a Cerebus acquisition system (Blackrock Microsystems), with 16-bit precision and a range of ± 8 mV.

To extract unit spiking activity, we re-referenced each electrode's signal offline by subtracting the mean signal of all the electrodes in the MEA, and then used a second order Butterworth filter to bandpass the signal between 0.3 and 3 kHz. Using a spike-sorting software package (Plexon Offline Sorter, Dallas, TX, USA), we identified spike waveforms by manually setting a negative or positive voltage threshold depending on the direction of putative action potentials. The voltage threshold was set to include noise signals used in calculating unit isolation quality (see below). Waveforms (duration, 1.067 ms; 32 samples per waveform) that crossed the voltage threshold were stored for spike sorting. Spike clusters were manually identified by viewing the first two principal components, and the difference in peak-to-trough voltage (voltage versus time) of the waveforms. We manually drew a boundary around clusters of waveforms that were differentiable from noise throughout the experimental session. In this manner, we identified a total of 989 putative single units across all sessions (average of 72 ± 21 units per participant). The average spike rate across all units was 2.82 ± .01 Hz. In addition to the spiking data, we also used a 500 Hz low pass filter to extract the LFP signals from each microelectrode, down-sampled to 1000 Hz, and then performed a similar line noise removal and channel selection procedure to that used for the iEEG channels to exclude artifacts related to epileptiform activity or other system level noise. Across the six participants, after pre-processing we retained recordings from 78 ± 27 MEA electrodes for further analysis.

## Single-unit recording quality measures

Due to variability in the signal quality across recordings and the subjective nature of spike sorting, we quantified the quality of each unit by calculating an isolation score and signal to noise ratio (SNR) (*Joshua et al., 2007*). The isolation score quantifies the distance between the spike and noise clusters in a 32-dimensional space, where each dimension corresponds to a sample in the spike waveform. The spike cluster consisted of all waveforms that were classified as belonging to that unit, and the noise cluster consisted of all waveforms that crossed the threshold that were not classified as belonging to any unit. The isolation score is normalized to be between 0 and 1, and serves as a measure to compare the isolation quality of all units across all experimental sessions and participants. Across participants, the mean isolation score for all units was .93 ± .1.

In addition to isolation quality, we computed the SNR for each unit:

$$SNR = \frac{V_{peak} - V_{trough}}{Noise * C}$$

where $V_{peak}$ and $V_{trough}$ are the maximum and minimum voltage values of the mean waveform, and $C$ is a scaling factor (set as 5). To obtain $Noise$, we subtracted the mean waveform from each individual waveform for each identified unit, concatenated these waveform residuals, and then computed the standard deviation of this long vector. Therefore, the noise term quantifies the within-unit variability in waveform shape. Across participants, the mean SNR for all units was 1.71 ± .12.

We estimated the instantaneous spike rate for each unit by convolving the spike rasters with a Gaussian kernel (σ = 25 ms). We used the mean and standard deviation of the spike rate over an entire experimental session to generate a z-scored spike rate for each unit.

## Ripple detection

We detected ripples in both the iEEG and LFP signals as previously reported (*Vaz et al., 2019*). We first bandpass filtered the voltage time series in the ripple band (80–120 Hz) using a second order

Butterworth filter, and then applied a Hilbert transform to extract the instantaneous amplitude and phase within that band. We selected events where the Hilbert envelope exceeded two standard deviations above the mean amplitude of the filtered traces. We only retained events that were at least 25 ms in duration and had a maximum amplitude greater than three standard deviations as ripples for analysis. We did not specify an upper limit for ripple duration. We joined adjacent ripples that were separated by less than 15 ms. We identified every ripple that satisfied these criteria in every electrode contact, and assigned each such identified ripple a start time index and an end time index. The difference between them defined the duration of each ripple.

To assess the overlap between detected ripples and inter-ictal epileptic discharge (IED) artifacts, we computed the joint probability of iEEG and LFP ripples and the identified IEDs for each participant. We found that IEDs overlapped with .79 ± .11% of iEEG ripples and with 1.38 ± .11% of LFP ripples across the six participants with MEAs (*Figure 1—figure supplement 5A,B*). We excluded all IEDs and high-frequency oscillations associated with IEDs (ripple on spike waveforms, pathologic ripples) and any detected ripple that overlapped with an IED from our analyses. The remaining ripples that we retained for our analyses therefore occurred without an associated IED and are more likely to be physiologic.

To examine the relation between ripple amplitude and spiking activity, as well as to examine the relation between ripples across spatial scales, we used the Hilbert phase and amplitude of the 80–120 Hz ripple band signal extracted from both the iEEG and LFP signals. To assess for a spectrum of ripple amplitudes and durations, we relaxed the detection thresholds to include all events during which the Hilbert amplitude of the LFP signal exceeded only one standard deviation above the mean amplitude of the filtered traces. We designated all such events with a minimum duration of 10 ms and with a maximum amplitude at least two standard deviations above the mean as putative ripples for these analyses.

To account for the possibility that ripples with higher amplitudes and therefore longer durations may be associated with more spiking activity by chance, we compared the true correlation between ripple amplitude and spiking activity to the correlations we would observe by chance. In each of 1000 permutations, we performed a random circular shift of the spike indices in each trial and computed the correlation between LFP ripple amplitude and spike rate across units and MEA electrodes. We compared the true correlation to the mean of the distribution of 1000 shuffled correlations in each participant. We determined that 1000 permutations was sufficient by initially examining the mean correlation as a function of the number of permutations in a single participant, and found that the mean value for the correlation observed by chance converged after only 500 permutations. We performed a similar permutation procedure when examining the relation between iEEG ripple amplitude and the proportion of active units, and between iEEG ripple amplitude and the number of underlying LFP ripples.

## Pairwise phase consistency

To examine the extent to which individual events such as spikes or ripples are aligned to consistent phases in the LFP or iEEG oscillations, we computed the pairwise phase consistency (PPC) (*Vinck et al., 2010*). Briefly, for each spike or ripple, we extracted the instantaneous phase of the LFP or iEEG signal either of individual frequencies or within low (2–10 Hz) or ripple band (80–120 Hz) frequency bands. For individual frequencies, we used the instantaneous phase extracted by convolving the LFP or iEEG time series with complex valued Morlet wavelets (wavelet number 6) for 60 frequencies logarithmically spaced between 2 and 400 Hz. To extract the instantaneous phase of the two frequency bands, 2–10 Hz and 80–120 Hz, we filtered the LFP and iEEG signal into each frequency band and then extracted the instantaneous phase from the complex time series generated by the Hilbert transform of the filtered time series. Across multiple spikes or ripples, we therefore generate a distribution of phases. To calculate the PPC, we computed the average angular distance, or vector dot product, for all pairs of phases in each distribution. We defined the preferred phase for each distribution as the phase angle of the complex mean of the distributions of these phases. In addition to PPC, we also assessed phase consistency by testing whether each distribution of phases significantly deviated from a uniform distribution using a Rayleigh test of uniformity.

We used PPC to examine the extent to which 80–120 Hz ripple band phases are aligned across all microelectrodes in each MEA during each ripple detected in the larger scale iEEG signal. In this

case, during every time point within each iEEG ripple, we collected a distribution of 80–120 Hz ripple band phases from all 96 microelectrodes, and computed the PPC on that distribution. We assigned the maximum PPC computed over the duration of each iEEG ripple as the microelectrode 80–120 Hz PPC for that iEEG ripple. In each participant, we then computed the correlation between iEEG ripple amplitude and 80–120 Hz PPC in the underlying LFP across all iEEG ripples identified from all retrieval trials. We compared these true correlations to chance using a shuffling procedure. In each of 100 permutations, we circularly shifted the time series of LFP phase by a random amount within each detected iEEG ripple and then computed the correlation between iEEG ripple amplitude and LFP PPC. We calculated the average correlation across permutations in each participant as the chance level. We performed an identical procedure when examining the extent to which the alignment of spiking activity to the 80–120 Hz ripple band signal in the LFP is correlated with the 80–120 Hz ripple band amplitude.

To examine the extent to which spiking activity is locked to individual frequencies in the LFP and iEEG signal, we computed PPC using the instantaneous phases of each spike from each unit. In each participant, we computed the average spike PPC across all units in each trial, and then computed the average across trials to generate a spike PPC value for each participant. In order to compare PPC values across participants, we converted the raw PPC to a z-score in each participant by using the mean and standard deviation of a null distribution of 100 spike PPC values generated by randomly shuffling the trial labels associated with the spike indices.

We then assessed whether the distribution of spike PPC values is significant across participants using a non-parametric cluster-based procedure. For each frequency, we compared the distribution of z-scored spike PPC values to zero using a t-test, thus generating a true t-statistic and p-value for each frequency. We then randomly permuted the participant-specific values by randomly reversing the sign of z-scored PPC within each participant and recomputing the average value of the distribution of permuted PPC values across participants. For n participants, this results in an empiric distribution of $2^n$ possible values that are all equally probable under the null hypothesis. We generated an empiric distribution from 1000 permutations for each frequency and calculated t-statistics for each of the permuted frequencies.

To correct for multiple comparisons across frequencies, we identified clusters of adjacent frequencies that exhibited a significant difference between the average PPC across participants and zero (where in each frequency cluster, p < .05). For each cluster of significant frequencies identified in the true and permuted cases, we defined a cluster statistic as the sum of the t-statistics within that frequency cluster. We retained the maximum cluster statistic during each of the 1000 permutations to create a distribution of maximum cluster statistics. We assigned p-values to each identified cluster of the true data by comparing its cluster statistic to the distribution of maximum cluster statistics from the permuted cases. We determined clusters to be significant and corrected for multiple frequency comparisons if their p-value calculated in this manner was less than .05.

We also compared spike PPC between two sets of conditions - PPC for spikes that occurred during an identified LFP ripple as compared to PPC for spikes that occurred outside an LFP ripple, and PPC for spikes that occurred during correct versus incorrect memory retrieval. We only included units for this analysis that exhibited a minimum of 10 spikes in each condition during an experimental session. In addition, because each condition tends to have a low total number of spikes in each trial, we computed PPC in these analyses by aggregating spiking events across trials rather than initially computing PPC within individual trials. Because we are making a direct comparison between PPC values within individual participants, we used the raw PPC rather than the z-scored value for these tests. In all cases, we computed the average PPC across all units separately for each condition in each participant. We then compared the average PPC between conditions by using a similar permutation procedure that corrects for multiple comparisons described above. In this case, in each permutation we randomly switched the label for each condition in each participant. To ensure that lower spike counts in one condition would not bias our results, we identified which condition had the lower total number of spikes, and randomly subsampled the spikes from the other condition. We performed this subsampling 200 times, computed PPC for each iteration, and assigned the average of the PPC from the 200 iterations of subsampling to the condition with the larger number of spikes. We repeated all of these analyses when examining the extent to which ripples are locked to individual frequencies, and to compare the extent of locking between conditions.

## Pairwise phase consistency of spiking

In order to obtain a measure of phase locking that does not depend on number of observations, we look at pairs of phases. Phases that are consistently clustered around a mean phase have a small angular distance to each other. The absolute angular distance is expressed as

$$d_f(\varphi_f, \omega_f) = |\varphi_f - \omega_f| mod \pi, \tag{1}$$

where $\varphi$ represents the phase of spike to a frequency bin and $\omega$ represents the phase of another spike from the same neuron to the same frequency bin. For each neuron, we can compute this for all frequency bins.

We compute the average pairwise circular distance (APCD), or the absolute angular distance between relative phases, which can be expressed as:

$$\hat{D} = \frac{2}{N(N-1)} \sum_{j=1}^{N-1} \sum_{k=j+1}^{N} d(\theta_j, \theta_k), \tag{2}$$

The pairwise phase consistency (PPC) is equivalent to the population statistic of the APCD, which is equivalent to the population statistic of the square of the phase-locking value.

We compute the sample estimate of the PPC by evaluating:

$$\hat{\gamma} = \frac{2}{N(N-1)} \sum_{j=1}^{N-1} \sum_{k=j+1}^{N} f(\theta_j, \theta_k), \tag{3}$$

where $f(\varphi, \omega) = cos(\varphi)cos(\omega) + sin(\varphi)sin(\omega)$ and N represents the number of spikes.

To efficiently compute the PPC of spikes to one frequency bin of the local field potential, we express each spike phase as a unit vector and evaluate the dot product for all pairs of unit vectors. We compute the spike-LFP PPC from the resulting symmetric matrix by removing the values along the diagonal and then taking the mean.

## Pairwise phase consistency of ripple oscillations

To measure the phase consistency of ripple oscillations across MEA electrodes, we compute the absolute angular distance using *Equation 3* where $\theta_j$ represent the phase of the ripple band signal for one MEA electrode, $\theta_k$ represents the phase of ripple band signal for a different MEA electrode for one time point, and N represents the number of MEA electrodes. Each time point within a iEEG ripple was treated as an observation for the MEA electrode. In other words, for a 50 ms long iEEG ripple, we evaluate the dot product for the pairs of ripple phases across all pairs of MEA electrodes. To efficiently compute the PPC of ripple oscillations across MEA electrode pairs, we express each ripple oscillation phase as a unit vector and compute the mean dot product for all pairs of unit vectors in a similar manner as spike-LFP PPC.

## MTL-ATL ripple cross-correlation

To measure the extent to which ripples in the anterior temporal lobe (ATL) are coupled with ripples in the medial temporal lobe (MTL), we identified the time index of peak ripple power for each rippled detected in both regions. We then generated cross-correlograms between MTL and ATL ripples (*Vaz et al., 2019*). For each electrode in the MTL, we computed a cross-correlogram with each electrode in the ATL. We then pooled these cross-correlograms across trials for each electrode pair in each participant. This generates a cross-correlogram for each pair of electrodes that we can compare between conditions and to a chance distribution (see below). To generate a single cross-correlogram representing the relation between the ATL and the MTL in each participant, we computed the average cross-correlogram across all electrode pairs.

For every pair of electrodes, we calculated a shift predictor for the cross-correlogram that characterizes the cross-correlation that would be expected by chance given the presentation of a stimulus (*Vaz et al., 2019*; *Brody, 1999*; *Steinmetz et al., 2000*; *Morris et al., 2004*). This chance distribution was generated by cross-correlating the time indices relative to the presentation of the stimulus for each ripple in an ATL electrode during an individual trial with the time indices of each ripple in an MTL electrode in every other trial. For *n* trials, we create *n* - 1 cross-correlations, which are then averaged to create a chance cross-correlogram (the shift predictor) for that trial. This procedure was repeated for all trials, and the average across all trials represents the average shift predictor for that

trial condition. We aggregated these chance cross-correlograms (shift predictors) across all electrode pairs that involve each region of interest to generate a shift predictor for each region.

The ratio between the true cross-correlogram and the shift predictor reflects the extent to which two signals are synchronized greater than would be expected by chance given the presentation of a stimulus. We calculated a normalized synchronization metric by finding the sum of the true cross-correlation values in a ± 50 ms window and then dividing by the corresponding area of the chance distribution. In this manner, our metric directly quantifies how much more synchronized the true case is relative to chance, which would result in a value of 1. To test the effect of a range of detection parameters on the correlation, we detected ripples using duration thresholds ranging from 10 to 40 ms, increasing in increments of 10 ms, and max amplitude thresholds ranging from 2 to 4 SD, increasing in increments of 1 SD. We used the same detection threshold for LFP and iEEG ripple detection. We used this metric to compare synchronization between detection parameters.

## LFP-iEEG ripple cross-correlation

To measure the coincidence of LFP and iEEG ripples, we identified the time index of peak ripple power for each rippled detected in each microelectrode (LFP ripple) and iEEG electrode (iEEG ripple). We then generated cross-correlograms between LFP and iEEG ripples. For each participant, we included four iEEG electrodes nearest to the MEA. To generate a single cross-correlogram representing the relation between the LFP and iEEG ripples in each participant, we computed the average across all electrode pairs. For every pair, we calculated chance cross-correlograms by randomly shifting in time each trial of the ripples detected in the microelectrode. We computed the average across trials for each electrode pair. We calculated a normalized synchronization metric by finding the average true cross-correlation values in a ± 50 ms window and then dividing by the corresponding area of the chance distribution. The ratio between the true and chance cross-correlograms quantifies how much more synchronous the LFP and iEEG ripples are relative to chance, with a value of 1 indicating a measurement equal to chance.

## Population spiking auto-correlation

To measure the extent to which units spike together in bursts within detected iEEG ripples, we summed the spiking across all units and computed the auto-correlogram of the population spiking within each detected iEEG ripple. We detected ripples using a duration threshold of 10 ms and an amplitude threshold of 1 SD with a maximum of at least 2 SD in four iEEG electrodes nearest to the MEA. To compare this auto-correlogram within ripples to spiking outside of ripples, we generated random duration matched windows between ripples and computed the chance population spiking auto-correlograms. We calculated a burst metric by finding the average of the true auto-correlogram in a ± 25 ms window centered around zero and then dividing by the corresponding area of the chance correlogram.

## Hartigan's test for bimodal distribution

To assess whether distributions of population spike rate, LFP ripple power and iEEG ripple power are bimodal, we used Hartigan's dip test. We postulated that these distributions would be bimodal if there were indeed transient bursts of activity and periods of little activity in between. The dip test computes the maximum difference between the empirical distribution function and the unimodal distribution function that minimizes that maximum difference (*Hartigan and Hartigan, 1985*). To compute the dip statistic, we generated a probability density function (PDF) of samples aggregated across all four second trials in 200 bins over the range of the data. We computed a true dip statistic for spike rate and for LFP ripple power for each microelectrode and for iEEG ripple power for each iEEG channel. We generated a chance distribution of dip statistics for unimodal distributions to quantify the significance of the true dip statistic. For this procedure, we randomly generated 10,000 uniform PDFs and z-scored the true dip statistic using the mean and standard deviation of the chance distribution. The average z-scored dip statistic across all microelectrodes was used for the spike rate and LFP ripple power for each participant. The average z-scored dip statistic across four iEEG channels in the anterior temporal lobe and iEEG channels in the medial temporal lobe were used to compute the z-scored dip statistic for each participant. This analysis was performed on the six participants with a MEA.

## Multiple oscillations detection algorithm detection of narrowband oscillations

We used an independent and previously validated method for detecting transient episodes of narrowband oscillations to assess whether ripples detected using duration and amplitude thresholds in the 80–120 Hz frequency range capture similar events detected using other approaches. For this procedure, we used the continuous-time wavelet transform (wavelet number 6) to compute the mean power spectrum over the trial, which is then used to generate a background 1/f fit. We generated a 1/f fit to the 70–200 Hz range of the power spectrum for each trial and identify narrowband oscillations that exceed it. The signal is then bandpass filtered within the identified narrowband frequency ranges and a Hilbert transform is used to compute the instantaneous power and phase. The instantaneous frequency is estimated using a frequency sliding estimation method previously described (*Cohen, 2014*). Periods in which the power is below the 1/f fit is removed. Given we perform this for each trial, we identify a unique narrowband oscillation for each trial for each iEEG electrode. For each participant, we aggregate the oscillations across trials across iEEG electrodes to generate a distribution of center frequencies of narrowband oscillations and a distribution of durations of the periods when the oscillations exceeds 1/f background signal.

## Meta analysis

Given the variability in number of ripples and other characteristics across participants, we quantified within and across participant variability and computed an estimate of the total true correlations. We assessed whether random variation accounts for the observed correlations by performing a meta-analysis where we used restricted maximum-likelihood estimation to fit a random effects model (*Viechtbauer, 2010*). For each participant, we computed the true correlation and z-scored it using a distribution of correlation values for shuffled data to generate the r equivalent, a measure of effect size. We computed the sampling variance for each participant from the number of samples (*Rosenthal and Rubin, 2003*). These measures were used to fit the random effects model.

## Acknowledgements

We thank J Chapeton, V Sreekumar, and Z Xie for helpful and insightful comments on the manuscript. We are indebted to all patients who have selflessly volunteered their time to participate in this study. This work was supported by the Intramural Research Program of the National Institute of Neurological Disorders and Stroke. This work was also supported by NINDS grant F31 NS113400 (APV). Conflicts of Interest: The authors declare no competing financial interests.

---

## Additional information

### Funding

| Funder | Grant reference number | Author |
|---|---|---|
| National Institute of Neurological Disorders and Stroke | F31 NS113400 | Alex P Vaz |
| National Institute of Neurological Disorders and Stroke | Intramural Research Program | Kareem A Zaghloul |

The funders had no role in study design, data collection and interpretation, or the decision to submit the work for publication.

### Author contributions

Ai Phuong S Tong, Conceptualization, Data curation, Formal analysis, Methodology, Validation, Visualization, Writing - original draft, Writing – review and editing; Alex P Vaz, Conceptualization, Data curation, Formal analysis, Funding acquisition, Methodology, Validation, Writing – review and editing; John H Wittig, Data curation, Methodology, Writing – review and editing; Sara K Inati, Data curation,

Writing – review and editing; Kareem A Zaghloul, Conceptualization, Data curation, Methodology, Project administration, Supervision, Writing – review and editing

**Author ORCIDs**
Ai Phuong S Tong http://orcid.org/0000-0002-2771-9504
John H Wittig http://orcid.org/0000-0003-0465-1022
Kareem A Zaghloul http://orcid.org/0000-0001-8575-3578

**Ethics**
Human subjects: Data were collected at the Clinical Center at the National Institutes of Health (NIH; Bethesda, MD). The Institutional Review Board (IRB) approved the research protocol (11-N-0051), and informed consent was obtained from the participants and their guardians.

**Decision letter and Author response**
Decision letter https://doi.org/10.7554/eLife.68401.sa1
Author response https://doi.org/10.7554/eLife.68401.sa2

## Additional files

### Supplementary files
• Transparent reporting form
• Source code 1. Matlab scripts to generate main figures.

### Data availability
Data and custom code used in this study can be found at https://doi.org/10.5061/dryad.5qfttdz6t. Source code and data files have been uploaded for Figures 1-4.

The following dataset was generated:

| Author(s) | Year | Dataset title | Dataset URL | Database and Identifier |
|---|---|---|---|---|
| Tong AP, Vaz A, Wiitig J, Inati S, Zaghloul K | 2021 | Ripples reflect a spectrum of synchronous spiking activity in human anterior temporal lobe | https://doi.org/10.5061/dryad.5qfttdz6t | Dryad Digital Repository, 10.5061/dryad.5qfttdz6t |

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
