## [Decision Letter]

**Decision letter after peer review:**

Thank you for submitting your article "Discrete ripples reflect a spectrum of synchronous spiking activity in human association cortex" for consideration by *eLife*. Your article has been reviewed by 3 peer reviewers, one of whom is a member of our Board of Reviewing Editors, and the evaluation has been overseen by Joshua Gold as the Senior Editor. The reviewers have opted to remain anonymous.

The reviewers have discussed their reviews with one another, and the Reviewing Editor has drafted this to help you prepare a revised submission. While the reviewers agreed the authors have an opportunity to contribute important findings related to memory, and high-frequency/spiking activity in humans they felt that the claims of ripples as discrete events was not supported. Further, they suggest increased transparency with regards to the possibility that these events may not be ripples but rather high frequency oscillatory events that reflect spiking activity. Further suggestions for revisions are below.

Essential revisions:

1) The results as-is do not support the claim that ripples are discrete events. The reviewers suggest the authors include additional analyses to demonstrate that ripples are discrete events or instead remove these claims and report on the findings/characterization of these high frequency events and their relationship to spiking activity and memory.

2) To unambiguously demonstrate that ripples are discrete events, the authors could generate a plot of the distribution of detection features, such as binned ripple power of the LFP electrodes, which would presumably need to be bimodal. If, as would be expected from the rodent literature and the results of this study, the authors instead observe a long-tailed distribution, the data would clearly demonstrate that these are not discrete events. If so, rather than fighting to prove the "discrete" nature of ripples, it would be more productive to embrace the variability in event sizes and take a more "signal detection" -style approach to this challenge. In line with this approach, methods have been developed which first model the noise of the signal and use this to set a somewhat less arbitrary threshold for event detection (Yu et al., *eLife* 2017), which demonstrate that events far smaller than would have been detected using traditional thresholds are likely to be biologically meaningful.

3) In addition to using more sensitive detection methods, the authors can ask how results, such as phase coupling or spiking modulation, vary over different sizes of events, and perhaps identify features that can inform less arbitrary detection thresholds for future work. Further, it would be extremely useful to see how the different sizes of LFP events are differentially detectable/reflected in the iEEG signal. For instance, it would be informative to know what proportion of events detected in the iEEG coincide with events in the LFP, and how this varies over a range of detection thresholds applied to each type of recording.

4) The title of Figure 1 would suggest that the authors will demonstrate the discrete nature of ripple events, however, the examples and analyses in this figure do not address this question. Unless the authors can show that there are clearly bimodal distributions underlying ripple events, It would be preferable to set aside the "discrete" claim throughout the manuscript and instead use this figure to address the detection methodology and show the characteristics of ripples in iEEG and LFP over a range of detection thresholds.

5) Also regarding Figure 1, it is unclear how much of the findings showing differential engagement of ripples/HFA on correct vs incorrect retrievals are re-prints of previously published findings, replication of previous findings with new or additional data, or new findings. Please specify.

6) An illustration like Figure 2A with much more detail and quantification would be extremely helpful for developing a better intuition for how to evaluate the resultant data. For instance, is a single iEEG contact consistently directly above, and exactly the same size as, the MEA, as suggested by Figure 2A? Where do the MEA electrode tips tend to lie in relation to cortical layers? Additionally, it would be critical to specify exactly how the spatial layout of electrodes, and the brain areas targeted, varied over the individual patients.

7) Regarding figures 2 and 3, more analyses of how often events detected in the iEEG would coincide with events detected in the LFP and/or cross-correlograms of the time of peak power for each event would be useful instead or in addition to the current approach of applying event detection in one data type and then plotting the continuous measure from the other datatype. In addition, it would be useful if the authors reported, in the text, the amount of variance for each relationship, not just whether a regression is significant. For example, in Figure 2c, the signal from a local LFP electrode captures ~36% of the variance in spiking rate, while the signal from the iEEG electrode captures ~1.4% of the variance. The authors are making claims about iEEG reflecting the LFP or spiking, and it's important that these claims be quantified with the strengths of the relationships, not just whether they are significant or not.

8) Regarding figures 2-4, would it be possible to glean more statistical power by capturing the variability of events within each patient as well as the variability across patients by using linear mixed-effects models for the summary statistics such as those shown in Figure 2D, F, H, etc? These models generally provide a much better way to describe relationships when large data samples have been collected from multiple subjects.

9) With regards to the quantification of the strengths of the relationships, it might be helpful if the authors quantified the same relationships across the spatial extent of the MEA. That might allow them to estimate the fall-off of predictability with distance on short length scales, which would give them a first order prediction of what they might see on the larger scales of the distances from the iEEG contacts to the MEA.

10) Additional raw data examples (similar to Figure 2B or 4A) including the full LFP band signal from several MEA sites (not only the 80-120Hz filtered traces) would be useful to add into the supplementary figures. In particular, it would be informative to be able to compare the full LFP signal to the full 1-200Hz iEEG trace.

11) In the experiment, authors assigned a randomly correct response time to 'pass' trials and categorized these 'pass' trials as incorrect. The randomly assigned response time smear out the effect of vocal-locked responses of incorrect trials. It might be one reason underlying the difference between incorrect and correct trials. Authors perhaps can include only the 'intrusive' trials as incorrect trials and see how the result look like.

12) The authors found that the amplitude of ripple events reflected the local spiking activities which was consistent with results from a study by Logothetis et al. showing that the gamma activity of LFP is related with local single unit activities. How could the authors distinguish the difference between their ripple activity and Logothetis's gamma activity?

13) What is the difference between iEEG ripples and LFP ripples? Do they differ in amplitude? Do they come from the same origin? I suggest adding details as to how they differ.

14) The micro arrays were placed within the expected resection area. I understand the potential damage to the brain if it is placed in a healthy area. However, how do the authors think it affects the results? It perhaps would be difficult to perform any control analyses given the current dataset. Maybe the authors can pay some lines in the discussion.

15) About the criterion of defining ripple events: In the methods part of 493-495, the criterion was already liberal compared to rodent studies. The authors however took an even more liberal criterion in another analysis (line 503-507). Perhaps the authors should call these events gamma bursts instead of ripples.

16) When performing the pairwise phase consistency analysis, they assigned the maximum PPC computed over the duration of each iEEG ripples as the LFP 80-120 Hz PPC for that iEEG ripple. It seems that they have a hypothesis that the PPC of LFP 80-120Hz which does not satisfy as ripple criterions is comparable with PPC of LFP ripples. I however did not find the supporting data.

17) I think the authors should be more transparent about the possibility that a portion of their defined "ripple" events may not in fact be ripples but high frequency activity events that relate to spiking activity. I suggest renaming these events to "ripple-like" and adding relevant discussion on this topic.

18) While the authors acknowledge the limitation of their ripple detection methods, they go on to argue in their conclusions that their results provide evidence that human ripple activity are thus more variable and exist on a "continuum of amplitudes and scales". I was not convinced by this argument since it is also possible that many of these events are simply not ripples and rather could be high frequency events that reflect spiking activity.

19) The statement "neural activity in the human cortex is organized into dynamic, discrete packets of information" seems out of place and not directly related to what the study is attempting to show. I suggest removing it or rephrasing.

20) Very difficult to evaluate Figure 2 since the legend is an exact copy from figure 1. Please revise.

21) It would be nice if the authors can include how many "ripple" events overlapped with IED events.

[Editors' note: further revisions were suggested prior to acceptance, as described below.]

Thank you for submitting your article "Ripples reflect a spectrum of synchronous spiking activity in human anterior temporal lobe" for consideration by *eLife*. Your article has been reviewed by 3 peer reviewers, one of whom is a member of our Board of Reviewing Editors, and the evaluation has been overseen by Joshua Gold as the Senior Editor. The reviewers have opted to remain anonymous.

Essential revisions:

1) Claims that ripples reflect discrete events should be removed or further analyses should be completed and results included to support this conclusion.

2) Simultaneous recording of LFP and iEEG, and the associated findings are a major strength of the study and should be further highlighted throughout the manuscript.

*Reviewer #1 (Recommendations for the authors):*

The manuscript is much improved and the authors have done a nice job responding to reviewer concerns. The added analyses, including the investigation of medial temporal and anterior temporal ripples as well as LFP-IEEG ripples are a nice addition. Further the authors have added important language acknowledging the variability in features of human ripples as reported and potential differences between those found in rodents. I think the community would benefit from the revised manuscript and recommend publication.

*Reviewer #2 (Recommendations for the authors):*

In their revised manuscript, Tong et al. make a solid case to link neural spiking with patterns of oscillatory activity recorded at the local field potential and intracranial EEG levels. They have softened their claims about discrete packets of activity, and instead focus on establishing relationships between the different neural recording approaches over a range of event sizes and detection thresholds. We have two major suggestions to improve the clarity of their manuscript.

First, while the authors have moved away from focusing on whether ripples can be considered discrete events, this claim still comes up in several prominent places (including in the titles of their first Results section and figure) and continues to cause confusion. The authors now claim equally or more strongly that ripples exist on a continuum, which seems contradictory and incompatible with the claim of discreteness. The claim of discreteness is not central to the important message that these events are meaningful, that they can be detected by the various recording modalities, and that they reflect underlying neural spiking activity. Given this, it would likely be best if the authors simply remove all claims of discreteness. However, we also appreciate that the authors have included distributions of ripple features and can show that these can appear bimodal. If the authors truly want to demonstrate discreteness, further explanation of Figure 2-Supplement 4 is needed as well as more prominent positioning of this result in a main figure and further analysis of the events detected by this method. Indeed, if this bimodality is truly robust, then an improved detection method would use the dip in the distribution as the detection threshold for each session/subject for all subsequent analysis. Further, this detection approach could/should then be compared to the more traditional thresholds.

Second, the authors note several times throughout the manuscript that a particular challenge in understanding ripples is the range and variety of detection methods used (for instance, lines 34-36). This is absolutely true, but a perhaps even more critical challenge is the difficulty of comparing ripples detected by the two major recording techniques – LFP for rodent work and iEEG for human work. Critically, the authors are particularly well-positioned to shed light and improve our understanding of how the LFP and iEEG signals relate to each other, and they should incorporate this point as a major strength of their study throughout their manuscript.

*Reviewer #3 (Recommendations for the authors):*

Thanks for authors' efforts on the revised manuscript. Questions proposed during the first round were carefully taken care of by authors. The manuscript was also properly revised.

---

## [Author Response]

Essential revisions:1) The results as-is do not support the claim that ripples are discrete events. The reviewers suggest the authors include additional analyses to demonstrate that ripples are discrete events or instead remove these claims and report on the findings/characterization of these high frequency events and their relationship to spiking activity and memory.

We thank the Reviewers for raising this important point and we agree that in our initial submission, there was some ambiguity as to whether the ripples we observe in our data are indeed discrete events. We completely agree with the Reviewers that, when considering ripples, there is a continuity of activity and that the central challenge for the field has been how to identify real events and separate them from others. To be clear, the purpose of our manuscript is not to claim that such a threshold exists. Our purpose instead was to offer evidence that even events that fall below arbitrarily defined thresholds still reflect underlying bursts of spiking activity, that these events are still punctate and temporally discrete, and that therefore these events are also likely functionally meaningful. We take as an analogy, for example, the case of a single spikes recorded from a neuron. Identifying some threshold that distinguishes the occurrence of a meaningful number of spikes from background activity, however, has been challenging. In a similar vein, our data suggest that there are indeed discrete ripple events, largely because these events are accompanied by underlying bursts of spiking activity and often are associated with discrete ripples in the MTL. The amplitude and strength of these events, however, can be quite variable, making it challenging to distinguish these events from baseline activity. We therefore do not claim to identify specific criteria for defining real events. As the Reviewers note, without such criteria we are left with the current situation where future studies will still need to use their own set of criteria. We completely agree. The purpose of our study, however, is to just highlight the point that using these arbitrarily defined criteria risks overlooking these other events that still may be meaningful for the brain.

In order to clarify this point and to address this concern more explicitly, we have added several changes to our manuscript and introduced several new analyses. First, as the Reviewers have suggested, we have tempered our claims that these are discrete events that represent separate packets of information. We acknowledge that there is certainly some variability in the size of these ripple events, that making a clean distinction between when these ripple events emerge as entities that are distinct from the background activity is challenging, and that we cannot be certain whether arbitrarily small events are functionally meaningful. We have revised our Introduction and Discussion accordingly to highlight these possibilities, and to discuss the larger point regarding the challenges in identifying these specific thresholds. We have also removed the claim that these are discrete ripples from the title of our manuscript.

Second, although we recognize that the data may not be conclusive, we have supplemented this discussion with additional analyses that, in our opinion, strongly suggest that these events are indeed transient in nature even when failing to meet previous thresholds. These include the following:

– We now present new analyses demonstrating that when ripples are detected in the iEEG recordings, we simultaneously detect bursts of spiking activity (Figure 2-S3). To characterize this, we evaluated the spike auto-correlogram within and outside of ripples and find a clear peak in the auto-correlogram, suggesting a burst of spiking activity, within ripples even when using a range of ripple detection parameters. We believe these data present strong evidence that the ripple events we identify are primarily characterized by underlying bursts of spiking activity, and that they therefore are indeed different from the background activity.

– We recognize the concern that there is some question as to whether ripples in the cortex should be considered the same as previously identified ripples in the MTL. We therefore reasoned that if we can demonstrate that cortical ripples are coupled with ripples in the MTL, this would provide evidence that, first, the cortical ripples may be functionally related to the previously established MTL ripples, and that, second, these cortical ripples indeed emerge as transient and discrete events since we would only observe coupling with MTL ripples if the cortical ripples are distinguishable from background cortical activity. We now introduce a new analysis to test this. We examine the relation between ripples detected in the anterior temporal lobe (the focus of our work) and ripples detected in the MTL (Figure 1-S4). Our data demonstrate a clear relation between cortical ripples and MTL ripples. We expand upon this point in the revised Discussion.

– We also directly address the Reviewer comments regarding the distribution of events, and the question as to whether this distribution is bimodal. First, using the same examples demonstrating the transient nature of the detected events, we generate a histogram of ripple band (80-120 Hz) power for both the iEEG and LFP, as well as a histogram of population spiking rate (Figure 2-S4). The examples illustrate the bimodal nature of our data. We formally test this using a statistical test of bimodality, the Hartigan dip test (described below in Essential Revision 2), and find that across participants, ripple band power in the iEEG and LFP and population spiking appear to be significantly bimodal.

– We now present several more examples of the filtered iEEG and LFP signals from different electrodes in different trials, while overlaying the population spiking activity (Figure 2-S4). These examples visualize the transient nature of these events, including both the ripples and the bursts of spiking activity. In a complementary analysis, we also visualized the average iEEG and LFP trace locked to each ripple event across a number of different electrodes (Figure 1-S1). These examples illustrate that the average detected event at both spatial scales is transient in nature.

– Finally, we used an independent and previously validated method for detecting transient episodes of narrowband oscillations, called the Multiple Oscillations Detection Algorithm (MODAL; Watrous et al., 2018, *eLife*). Briefly, this algorithm identifies time points during which narrowband oscillations exceed the background 1/f noise in a signal. Without specifying our ripple band of interest (80-120 Hz), this algorithm identified discrete time periods in which narrowband oscillations with center frequencies within the 80-120 Hz ripple band exceeded the background power (Figure 1-S3). These time periods overlapped with the ripple events we identified using our rippled detection approach, providing further evidence that the events we identified as cortical ripples are transient and discrete in nature.

Together, we believe these additional analyses (described in greater detail below) provide stronger support to the possibility that the identified cortical ripples in our recordings are transient and discrete events that reflect underlying bursts of spiking activity. As noted above, we also acknowledge that these events more likely exist on a continuum, and that there is not always a clear separation between what constitutes one ripple event and the background activity. We think this is a fertile area of investigation and that there are several points to consider regarding this question, which we now address in the revised Discussion.

2) To unambiguously demonstrate that ripples are discrete events, the authors could generate a plot of the distribution of detection features, such as binned ripple power of the LFP electrodes, which would presumably need to be bimodal. If, as would be expected from the rodent literature and the results of this study, the authors instead observe a long-tailed distribution, the data would clearly demonstrate that these are not discrete events. If so, rather than fighting to prove the "discrete" nature of ripples, it would be more productive to embrace the variability in event sizes and take a more "signal detection" -style approach to this challenge. In line with this approach, methods have been developed which first model the noise of the signal and use this to set a somewhat less arbitrary threshold for event detection (Yu et al., eLife 2017), which demonstrate that events far smaller than would have been detected using traditional thresholds are likely to be biologically meaningful.

We thank the Reviewer for this suggestion. We have adopted two approaches to address this concern. First, as the Reviewer suggests, we do indeed embrace the variability in event sizes for the ripples we detect. Indeed, in many ways, this is the purpose of our study, to demonstrate that these events do take on a variety of amplitudes and durations, and that the extent to which we observe increases in ripple amplitude or duration reflects the extent of underlying synchronized spiking activity (and synchronized micro-scale ripple activity).

However, we also specifically examine the question as to whether the distribution of ripple band features is bimodal. We plot the distributions of ripple band amplitude, as the Reviewer has suggested, and population spiking rate (Figure 2-S4). We then test whether these distributions are bimodal using an established test of bimodality, the Hartigan’s dip test (Hartigan and Hartigan, 1985, Annals of Statistics). This test computes the maximum difference between the empirical distribution function and the unimodal distribution function that minimizes that maximum difference (z-scored against a distribution of randomly generated unimodal distributions for each electrode). We use the dip test to assess whether the population spike rate, iEEG ripple power, and LFP ripple power distributions are bimodal for each electrode. The dip test for the bimodality of population spiking is significant in and across all participants (6.802 +/- 1.013 z). In addition, we find the test for the bimodality of iEEG and LFP ripple amplitude to be significant across participants, and to be significant within individual participants in at least four of the six participants with MEAs (LFP: 4.323 +/- 1.257 z; iEEG: 3.236 +/- 0.669 z).

To complement this analysis, we have also introduced additional analyses as noted above. These include (1) an analysis demonstrating that the identified ripple events are characterized by underlying bursts of spiking activity that is more auto-correlated within than outside of ripple events (Figure 2-S3), (2) an analysis demonstrating that the ripple events detected in the anterior temporal lobe are strongly associated with discrete ripple events in the MTL (Figure 1-S4), and (3) an analysis using an independent and validated approach for detecting transient periods of narrowband oscillations (Multiple Oscillations Detection Algorithm; MODAL; Watrous et al., 2018, *eLife*) that we show similarly identifies transient ripple band events in our data (Figure 1-S3). Together, we believe that these new analyses provide additional evidence supporting our interpretation that these cortical ripples are transient and discrete events of variable size.

3) In addition to using more sensitive detection methods, the authors can ask how results, such as phase coupling or spiking modulation, vary over different sizes of events, and perhaps identify features that can inform less arbitrary detection thresholds for future work. Further, it would be extremely useful to see how the different sizes of LFP events are differentially detectable/reflected in the iEEG signal. For instance, it would be informative to know what proportion of events detected in the iEEG coincide with events in the LFP, and how this varies over a range of detection thresholds applied to each type of recording.

We think these are good suggestions and have now performed these analyses in our revised manuscript. First, as the Reviewer has suggested, we have now computed the extent to which spiking activity is locked to high frequency oscillations in the LFP by measuring the spike pairwise phase consistency (PPC) and examined how this is related to the LFP ripple amplitude. In our original analysis, we had demonstrated a significant relation between spike-LFP PPC and LFP ripple band amplitude during ripples as compared to the time periods outside of ripples (Figure 4G). We now examine how spike-LFP PPC vary with duration and amplitude of events that are detected as we change the detection parameters. For every change in detection parameter (duration, amplitude) we quantified the spike-LFP PPC of spikes within ripples. We found that spike-LFP PPC increases with increasing amplitude and duration thresholds (Figure 4-S2).

Second, as the Reviewer suggests, we also examine the proportion of LFP events that coincide with iEEG events, and how this proportion is affected over a range of detection thresholds (Figure 3-S2). We vary the maximum amplitude threshold from 2 SD to 4 SD and the duration threshold from 10 ms to 40 ms. For each set of detection parameters, we measure the cross-correlation between the detected LFP and iEEG ripples. We find that LFP and iEEG ripples co-occur above chance across patients for all detection thresholds.

4) The title of Figure 1 would suggest that the authors will demonstrate the discrete nature of ripple events, however, the examples and analyses in this figure do not address this question. Unless the authors can show that there are clearly bimodal distributions underlying ripple events, It would be preferable to set aside the "discrete" claim throughout the manuscript and instead use this figure to address the detection methodology and show the characteristics of ripples in iEEG and LFP over a range of detection thresholds.

We appreciate the Reviewer’s suggestion to set aside the claim that the cortical ripples we detect are discrete events. As we have noted above, we now acknowledge this point that these events likely exist on a continuum of activity. We have revised our Discussion to highlight this possibility more clearly. In addition, we have provided additional analyses as noted above that provide additional evidence supporting the possibility that the detected events are transient and discrete in nature.

As the Reviewer has noted here, the detection of these ripple events is highly dependent upon the parameters used for detection (e.g., amplitude and duration thresholds). We have therefore embraced the Reviewer’s suggestion to characterize ripples we detect in the iEEG and LFP signals over a range of detection thresholds. In our revised manuscript, we now present the distribution and descriptive statistics for ripple amplitude and duration when we use maximum amplitude thresholds ranging from 2 to 4 SD, in increments of 0.25, and duration thresholds ranging from 10 ms to 40 ms, in increments of 5 ms, for detecting ripples (Figure 1-S2). As expected, the number of detected iEEG and LFP ripples decreases as we increase these thresholds.

5) Also regarding Figure 1, it is unclear how much of the findings showing differential engagement of ripples/HFA on correct vs incorrect retrievals are re-prints of previously published findings, replication of previous findings with new or additional data, or new findings. Please specify.

We thank the Reviewer for raising this question and agree that this point was not clear in our original submission. High frequency activity changes associated with correct memory retrieval have been shown before in previously published studies, as the Reviewer notes. Here, we replicate these previous findings but with new data. More importantly, however, we use this initial figure to demonstrate that the changes in high frequency activity can be explained by an increase in the number of transient ripple events, which has not been explicitly demonstrated before. We have now clarified this point in the revised Methods.

6) An illustration like Figure 2A with much more detail and quantification would be extremely helpful for developing a better intuition for how to evaluate the resultant data. For instance, is a single iEEG contact consistently directly above, and exactly the same size as, the MEA, as suggested by Figure 2A? Where do the MEA electrode tips tend to lie in relation to cortical layers? Additionally, it would be critical to specify exactly how the spatial layout of electrodes, and the brain areas targeted, varied over the individual patients.

We agree with the Reviewer that providing more detail regarding the relative positions of the iEEG electrodes and MEA would be informative. As the Reviewer notes, our initial schematic in Figure 2A would suggest that the MEA sits directly below a single iEEG contact. In a single participant, this was indeed the case, and we have now added this information to our revised version of Figure 2. However, the relative placement of the MEA is variable across participants. We have now added a Supplementary figure (Figure 2-S1) that documents the position of each MEA with respect to the four nearest iEEG electrodes in each participant and with respect to the anterior temporal lobe. We have also provided information in our revised Methods about the length of each MEA electrode (1 mm), and that while we do not have specific information about the cortical layers from which we are recording, we estimate the tips to lie within layers 3 or 4.

7) Regarding figures 2 and 3, more analyses of how often events detected in the iEEG would coincide with events detected in the LFP and/or cross-correlograms of the time of peak power for each event would be useful instead or in addition to the current approach of applying event detection in one data type and then plotting the continuous measure from the other datatype. In addition, it would be useful if the authors reported, in the text, the amount of variance for each relationship, not just whether a regression is significant. For example, in Figure 2c, the signal from a local LFP electrode captures ~36% of the variance in spiking rate, while the signal from the iEEG electrode captures ~1.4% of the variance. The authors are making claims about iEEG reflecting the LFP or spiking, and it's important that these claims be quantified with the strengths of the relationships, not just whether they are significant or not.

We thank the Reviewer for this suggestion and have now explicitly examined the cross-correlograms between the detected iEEG and LFP ripples. We have now included these cross-correlograms in Figure 3-S2. We have also examined how these cross-correlograms change as we change the ripple detection criteria and found that over a wide range of amplitude and duration thresholds, we still observe a significant relation between iEEG and LFP ripples.

We also agree with the Reviewer that it would be important and helpful to explicitly demonstrate the amount of variance we observe in each relationship we examine. To address this, we have now performed a meta-analysis on the correlations that we observe between LFP ripple amplitude and spiking, between iEEG ripple amplitude and percent spiking units, and between iEEG ripple amplitude and LFP ripple pairwise phase consistency (PPC) across participants. In each participant, we compared the observed correlation to a distribution of correlation values for shuffled data. We used this to compute both the *r* equivalent in order to generate a measure of the effect size and the sampling variance for each participant (Rosenthal and Rubin, 2003, Psychology Methods). Given the variability in number of ripples and other characteristics of the samples across participants, we assessed how within or across participant variability affected the true effects. We found that the average effect size is significantly greater than zero across participants for (1) LFP ripple amplitude and spiking (r = 0.055 [0.01, 0.10], Z = 2.449, p = 0.014), (2) iEEG ripple amplitude and percentage of spiking units (r = 0.037 [0.01, 0.06], Z = 3.097, p = 0.002), and (3) iEEG ripple amplitude and LFP ripple PPC (r = 0.034 [0.01, 0.06], Z = 3.13, p = 0.002). We now present the results of these new analyses in Figure 2 and Figure 3.

8) Regarding figures 2-4, would it be possible to glean more statistical power by capturing the variability of events within each patient as well as the variability across patients by using linear mixed-effects models for the summary statistics such as those shown in Figure 2D, F, H, etc? These models generally provide a much better way to describe relationships when large data samples have been collected from multiple subjects.

We agree with the Reviewer that introducing additional analyses capturing the variability across participants would be informative. To that end, as described above, we have now performed a meta-analysis across participants using the correlations we observe between LFP ripple amplitude and spiking, between iEEG ripple amplitude and percent spiking units, and between iEEG ripple amplitude and LFP ripple pairwise phase consistency (PPC) across participants. In this meta-analysis, we used a restricted maximum-likelihood estimation to fit a random effects model (Viechtbauer, 2010, Journal of Statistical Software). This new analysis demonstrates the effect size within each participant, while also demonstrating that these effects are significant across participants.

9) With regards to the quantification of the strengths of the relationships, it might be helpful if the authors quantified the same relationships across the spatial extent of the MEA. That might allow them to estimate the fall-off of predictability with distance on short length scales, which would give them a first order prediction of what they might see on the larger scales of the distances from the iEEG contacts to the MEA.

This is also a good suggestion. As noted above, we have now explicitly examined the cross-correlation between ripple events detected in the iEEG and in the LFP in order to establish that the coupling we observe is robust despite using a variety of different detection criteria. In addition, we have now also examined how this relation changes as a function of Euclidian distance between the MEA and the four closest iEEG contacts (Figure 3). Across participants, we find a significant negative correlation between distance and the correlation between iEEG and LFP ripples (r = -0.572, p = 0.0035; mean +/- SEM, r = -0.595 +/- 0.271, n = 6; t(5) = – 2.40, p = 0.0615). In addition, in one participant, we divided the MEA into smaller groups of micro-electrodes, and computed the average cross-correlogram between the iEEG ripples in nearby iEEG electrodes and the LFP ripples within each subgroup of MEA micro-electrodes. We find that these correlations are stronger in the subgroup of micro-electrodes that are physically closer to the iEEG electrode. We now present these results in Figure 3-S3.

10) Additional raw data examples (similar to Figure 2B or 4A) including the full LFP band signal from several MEA sites (not only the 80-120Hz filtered traces) would be useful to add into the supplementary figures. In particular, it would be informative to be able to compare the full LFP signal to the full 1-200Hz iEEG trace.

We also agree that this is a good suggestion. We have now included an example of the unfiltered 1-200 Hz LFP signals for several MEA electrodes and the simultaneous 1-200 Hz iEEG signal from a nearby iEEG electrode during a four second period (Figure 2-S2). This example is intended to allow for a comparison of the full signals, as recommended by the Reviewers.

11) In the experiment, authors assigned a randomly correct response time to 'pass' trials and categorized these 'pass' trials as incorrect. The randomly assigned response time smear out the effect of vocal-locked responses of incorrect trials. It might be one reason underlying the difference between incorrect and correct trials. Authors perhaps can include only the 'intrusive' trials as incorrect trials and see how the result look like.

We thank the Reviewer for making this point. In our original manuscript, we only considered incorrect trials as those in which the participant either made an intrusion or vocalized the word ‘pass’. As the Reviewer notes, there are other trials in which the participant did not respond at all, but we did not include these trials in our analysis for the exact reason that the Reviewer highlights, which is to ensure that vocalization itself is not introducing an artificial difference between correct and incorrect trials. We apologize that this was not clear in our manuscript, and we have now revised the Methods to make this point more clear.

12) The authors found that the amplitude of ripple events reflected the local spiking activities which was consistent with results from a study by Logothetis et al. showing that the gamma activity of LFP is related with local single unit activities. How could the authors distinguish the difference between their ripple activity and Logothetis's gamma activity?

We thank the Reviewer for raising this point, and we agree that the distinction between ripples and gamma activity as described by Logothetis’ group would be important to clarify. In our assessment of the studies published by Logothetis and colleagues, gamma oscillations are specified as >50 Hz (Panagiotaropoulos et al., 2012, Neuron) and 30-90 Hz (Berens et al., 2008, Fron. Systems Neuroscience). The studies find a correlation between spiking and gamma oscillations, and the correlation varies depending on stimulus features. We similarly find a strong relation between spiking activity and ripples which in our analyses occupy a narrow band of frequencies between 80-120 Hz. This narrow band of activity falls within the broader range of frequencies identified by Logothetis, and so it is possible that the phenomena are related and that the gamma band activity described by Logothetis and colleagues simply includes this narrow ripple band. Of note, we find that spiking activity in our data is locked to this narrow band, suggesting that this frequency band reflects the burst of synchronized spiking. We have now discussed this point and relation with previous studies examining the relation between spiking activity and gamma band activity in the revised Discussion.

13) What is the difference between iEEG ripples and LFP ripples? Do they differ in amplitude? Do they come from the same origin? I suggest adding details as to how they differ.

We thank the Reviewer for raising this question. We refer to iEEG ripples as ripples that we capture at a relatively larger spatial scale using the clinical iEEG electrode contacts. We refer to LFP ripples as ripples that we capture at the micro-scale through the micro-electrodes in the MEA. We find that the amplitude of the iEEG ripples (at the larger scale) reflects the number and synchrony of underlying LFP ripples (at the smaller scale). The LFP ripples in turn are related to the underlying spiking activity. Together, these data therefore suggest that the ripples come from the same origin, underling spiking activity, and simply reflect the aggregation of that activity across different spatial scales. We have now made this relation more clear by introducing an analysis examining the correlation in ripple activity across the two spatial scales (Figure 3 – S2) and have also provided an example of the raw LFP and iEEG signals for comparison (Figure 2 – S2).

14) The micro arrays were placed within the expected resection area. I understand the potential damage to the brain if it is placed in a healthy area. However, how do the authors think it affects the results? It perhaps would be difficult to perform any control analyses given the current dataset. Maybe the authors can pay some lines in the discussion.

This is also a good point and we thank the Reviewer for this suggestion. The micro-electrode arrays are placed in an area of the brain that is ultimately resected, and therefore may contain tissue affected by the patient’s epilepsy. We do not believe this has an appreciable effect on our results because we apply several preprocessing steps to remove epileptic activity from the signals. To address this point further, we have now performed additional analyses examining this issue and found that only ~1% of IEDs were also detected as ripples, suggesting that we are primarily studying physiological signals that would have otherwise been recorded in healthy tissue (Figure 1 – S5). In addition, the significant differences in ripple rates that we observe related to behavior would also suggest that we are recording physiological signals. As the Reviewer points out, it is difficult to perform control analyses to ensure that our findings are not contaminated by activity related to the patients’ epilepsy. However, in light of this concern, we have now also commented on this point in the revised Discussion.

15) About the criterion of defining ripple events: In the methods part of 493-495, the criterion was already liberal compared to rodent studies. The authors however took an even more liberal criterion in another analysis (line 503-507). Perhaps the authors should call these events gamma bursts instead of ripples.

The Reviewer is correct in that we use a more liberal threshold in some of our analyses, and that in some cases, we are detecting relatively brief bursts of activity within the gamma band. However, this was intentional as we do not believe there is an absolute set of criteria for defining what is and what is not a ripple. Instead, we believe our data demonstrate that these events exist on a continuum. The motivation for using the more liberal threshold in some of our analyses was to demonstrate that even when ripples exhibit lower amplitudes or durations, they still reflect underlying spiking activity. Therefore, even ripples that do not meet previously established amplitude or duration criteria may still be functionally relevant. We have now revised our Introduction and Discussion to make this point more clear, which we believe will make the motivation for using more liberal thresholds in some of our analyses more apparent.

16) When performing the pairwise phase consistency analysis, they assigned the maximum PPC computed over the duration of each iEEG ripples as the LFP 80-120 Hz PPC for that iEEG ripple. It seems that they have a hypothesis that the PPC of LFP 80-120Hz which does not satisfy as ripple criterions is comparable with PPC of LFP ripples. I however did not find the supporting data.

We agree with the Reviewer that in our original manuscript, by including all LFP signals when computing the PPC over the duration of each iEEG ripple, the computation of the PPC therefore may include LFP signals that do not satisfy our criteria for ripples. In order to address this concern, we have now repeated this analysis, using in this case only microelectrodes with detected ripples to compute the PPC during each iEEG ripple and find that the results are unchanged. The mean +/- SEM correlation across participants using only microelectrodes with detected ripples is 0.066 +/- 0.020, representing a significant correlation when analyzed across participants (p = .014). There is not a significant difference compared to when we looked at all micro-electrodes (t(5) = -0.63, p = 0.56).

17) I think the authors should be more transparent about the possibility that a portion of their defined "ripple" events may not in fact be ripples but high frequency activity events that relate to spiking activity. I suggest renaming these events to "ripple-like" and adding relevant discussion on this topic.

We think the Reviewer is highlighting an important point, which is how to define a ripple. As the Reviewer is aware, ripples have been primarily studied in the rodent MTL and therefore have taken on a very specific meaning. Whether similar phenomenon in other brain regions and in other species should also be called ripples has been a matter of intense debate. Our data do not attempt to resolve this debate, but instead highlight the point that in the human cortex, we observe punctate bursts of narrow band 80-120 Hz activity that are strongly associated with underlying spiking activity. In many ways, this looks very similar to what one would expect to find in a hippocampal ripple, but it is certainly possible that these events are only ‘ripple-like’. We agree with the Reviewer that this issue is not settled by our data. We have now highlighted this point in the revised Discussion. For the purposes of simplicity, we still refer to the events studied here as ripples, but we make clear in the revised Introduction and Discussion that our data do not establish that these events are identical to ripples traditionally recorded in the rodent MTL.

18) While the authors acknowledge the limitation of their ripple detection methods, they go on to argue in their conclusions that their results provide evidence that human ripple activity are thus more variable and exist on a "continuum of amplitudes and scales". I was not convinced by this argument since it is also possible that many of these events are simply not ripples and rather could be high frequency events that reflect spiking activity.

We agree with the Reviewer that the distinction between ripples and high frequency events that reflect spiking activity is unclear. This is related to how one defines what constitutes a ripple, as discussed above. However, we also note that previous studies examining the relation between high frequency activity and spiking activity have largely established this relation when using a relatively broad range of frequencies. Here we find that this relation is specific to a relatively narrow band, the 80-120 Hz ripple band. Our selection for this frequency band is based on a number of prior studies that use human intracranial electrode recordings and magnetoencephalography and report ripples as high frequency oscillations with center frequencies around 100 Hz (Bragin et al., 1999, Epilepsia; Staba et al., 2002, J Neurophysiology; Staresina et al., 2015, Nature Neuroscience; Ramirez-Villegas et al., 2015, PNAS; Vaz et al., 2019, Science; Norman et al., 2019, Science; Liu et al., 2019; Cell; Ngo et al., 2020, *eLife;* Norman et al., 2021, Neuron). To address this point further, we have now introduced an additional analysis using a previously validated approach for detecting narrowband oscillations called the Multiple Oscillations Detection Algorithm, or MODAL (Watrous et al., 2018, *eLife*). Briefly, this approach identifies all transient events that exceed the 1/f background noise for each electrode. Notably, this approach is agnostic to the frequency band of interest – we provide the broadband high frequency signal, 70-200 Hz, and the MODAL method extracts the narrow band oscillations that exceed this 1/f background signal. The MODAL method identified discrete periods in which narrow band oscillations exceeded background noise, and these events overwhelmingly had center frequencies in the 80-120 Hz ripple band. The average center frequency of the identified events was 87.3 +/- 3.5 Hz (mean +/- SD). The average duration of the events detected by the MODAL method was 33.5 +/- 3.0 ms, compared to a duration of 32.8 +/- 6.3 ms for the ripples that we detected using our standard approach. We now present these results as a supplementary point (Figure 1 – S3). These data therefore suggest that many of the high frequency events captured in human intracranial recordings are narrow band events that fall within this 80-120 Hz ripple band.

19) The statement "neural activity in the human cortex is organized into dynamic, discrete packets of information" seems out of place and not directly related to what the study is attempting to show. I suggest removing it or rephrasing.

We agree with the Reviewer that our data do not provide direct evidence that the ripple events we identify reflect packets of information. We have now revised this statement accordingly to temper that claim.

20) Very difficult to evaluate Figure 2 since the legend is an exact copy from figure 1. Please revise.

We apologize for this error. We have now corrected the legend for Figure 2.

21) It would be nice if the authors can include how many "ripple" events overlapped with IED events.

We thank the Reviewer for this suggestion and we have now computed the joint probability of iEEG and LFP ripples and interictal epileptic discharges (IEDs) for each participant. We found that IEDs overlapped with 0.79 +/- 0.11 % of iEEG ripples and with 1.38 +/- 0.11 % of LFP ripples, reported as mean +/- SEM across six participants. We now illustrate this overlap in a new supplementary figure that demonstrates two example trials of how the overlap between ripple events and IED events appears in the recordings, and a raster plot across many trials for an example iEEG electrode (Figure 1 – S5). Although we retain these ripples for this supplementary analysis, we have also clarified that we only retain the ripples for our main analyses in our study that do not co-occur with IED events.

[Editors' note: further revisions were suggested prior to acceptance, as described below.]

Reviewer #2 (Recommendations for the authors):In their revised manuscript, Tong et al. make a solid case to link neural spiking with patterns of oscillatory activity recorded at the local field potential and intracranial EEG levels. They have softened their claims about discrete packets of activity, and instead focus on establishing relationships between the different neural recording approaches over a range of event sizes and detection thresholds. We have two major suggestions to improve the clarity of their manuscript.First, while the authors have moved away from focusing on whether ripples can be considered discrete events, this claim still comes up in several prominent places (including in the titles of their first Results section and figure) and continues to cause confusion. The authors now claim equally or more strongly that ripples exist on a continuum, which seems contradictory and incompatible with the claim of discreteness. The claim of discreteness is not central to the important message that these events are meaningful, that they can be detected by the various recording modalities, and that they reflect underlying neural spiking activity. Given this, it would likely be best if the authors simply remove all claims of discreteness. However, we also appreciate that the authors have included distributions of ripple features and can show that these can appear bimodal. If the authors truly want to demonstrate discreteness, further explanation of Figure 2-Supplement 4 is needed as well as more prominent positioning of this result in a main figure and further analysis of the events detected by this method. Indeed, if this bimodality is truly robust, then an improved detection method would use the dip in the distribution as the detection threshold for each session/subject for all subsequent analysis. Further, this detection approach could/should then be compared to the more traditional thresholds.

We thank the Reviewer for raising this point, and we agree that we should soften our claim that these ripples reflect discrete events. We agree that the claim of discreteness is not central to the important message of our manuscript. We have now carefully reviewed our manuscript and removed all claims of discreteness.

Second, the authors note several times throughout the manuscript that a particular challenge in understanding ripples is the range and variety of detection methods used (for instance, lines 34-36). This is absolutely true, but a perhaps even more critical challenge is the difficulty of comparing ripples detected by the two major recording techniques – LFP for rodent work and iEEG for human work. Critically, the authors are particularly well-positioned to shed light and improve our understanding of how the LFP and iEEG signals relate to each other, and they should incorporate this point as a major strength of their study throughout their manuscript.

We agree with the Reviewer and we have emphasized in our revised Discussion, but also throughout the manuscript, that our data are well positioned to examine ripples across two different recording scales, LFP and iEEG.